# DistRL: An Asynchronous Distributed Reinforcement Learning Framework for On-Device Control Agents

**Taiyi Wang**[1,2*†], **Zhihao Wu**[3*], **Jianheng Liu**[4], **Jianye Hao**[3,6], **Jun Wang**[5], **Kun Shao**[3†]
[1]University of Cambridge, [2]Powersense Technology Limited, [3]Huawei Noah's Ark Lab,
[4]University College London, [5]AI Centre, University College London, [6]Tianjin University

## Abstract

On-device control agents, especially on mobile devices, are responsible for operating mobile devices to fulfill users' requests, enabling seamless and intuitive interactions. Integrating Multimodal Large Language Models (MLLMs) into these agents enhances their ability to understand and execute complex commands, thereby improving user experience. However, fine-tuning MLLMs for on-device control presents significant challenges due to limited data availability and inefficient online training processes. This paper introduces DistRL, a novel framework designed to enhance the efficiency of online RL fine-tuning for mobile device control agents. DistRL employs centralized training and decentralized data acquisition to ensure efficient fine-tuning in the context of dynamic online interactions. Additionally, the framework is backed by our tailor-made RL algorithm, which effectively balances exploration with the prioritized utilization of collected data to ensure stable and robust training. Our experiments show that, on average, DistRL delivers a $3\times$ improvement in training efficiency and enables training data collection $2.4\times$ faster than the leading synchronous multi-machine methods. Notably, after training, DistRL achieves a **20%** relative improvement in success rate compared to state-of-the-art methods on general Android tasks from an open benchmark, significantly outperforming existing approaches while maintaining the same training time. These results validate DistRL as a scalable and efficient solution, offering substantial improvements in both training efficiency and agent performance for real-world, in-the-wild device control tasks. The code is available at https://github.com/ai-agents-2030/DistRL-open.

## 1 Introduction

The integration of Large Language Models (LLMs) into agents capable of complex tasks has gained momentum with initiatives like AutoGPT (Yang et al., 2023a), HuggingGPT (Shen et al., 2024), and MetaGPT (Hong et al., 2023), etc. These agents extend beyond language processing to perform sophisticated functions, leveraging their reasoning abilities to interact with and manipulate environments effectively.

One of the key factors driving this trend is the advent of Multimodal Large Language Models (MLLMs), which can process diverse inputs such as text, images, audio, and video, thereby significantly expanding the scope of LLM applications (Alayrac et al., 2022; Achiam et al., 2023; Zheng et al., 2024; Li et al., 2023). This versatility also enables MLLM-based on-device control agents—intelligent systems embedded within mobile devices that manage and operate applications to execute user commands seamlessly—to interact more naturally and efficiently with their surroundings, completing more complex tasks that require a deeper understanding of context and the ability to learn from interactions. For instance, agents designed to operate smartphone applications can interpret screenshots from the operating system, demonstrating flexibility and adaptability that make them valuable tools in a wide range of scenarios (Yang et al., 2023b; Wang et al., 2024a; Christianos et al., 2024; Papoudakis et al., 2025). These agents are essential for tasks such as automating

---

*Equal Contribution.
†Corresponding Email: Taiyi.Wang@cl.cam.ac.uk, shaokun2@huawei.com

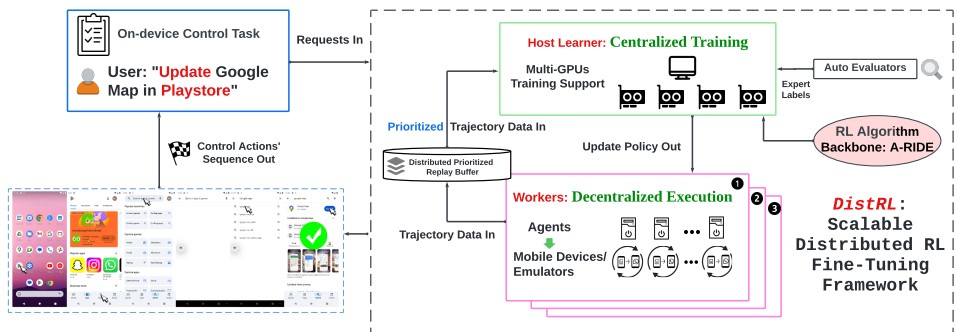

**Figure 1:** Overview of On-device LLM control with DISTRL.

app interactions, managing settings, and enhancing user productivity by providing intuitive control over device functionalities.

Nonetheless, a gap remains between LLMs' general reasoning capabilities and their effectiveness in GUI-based device control. While LLMs can process information, they struggle with rational behavior and error recovery in real-world environments. Consequently, prior work in building on-device agents often relies on constructing complex wrappers, combining them with prompting, search, or tool use. Without updating the model's weights, the effectiveness of these agents remains inherently limited by the capabilities of the base model (Bai et al., 2022). Moreover, MLLMs are prone to confidently generating incorrect content that deviates from desired outputs (Ouyang et al., 2022; Ziegler et al., 2019; Bai et al., 2022). To address these challenges, introducing reinforcement learning (RL)-based fine-tuning methods, such as Reinforcement Learning from AI Feedback (RLAIF) (Yu et al., 2024; Lee et al., 2023), which evolved from Reinforcement Learning from Human Feedback(RLHF), becomes essential. RLAIF leverages AI-generated feedback to align model outputs with intended behaviors and performance criteria. By incorporating RL-based methods, models can learn to optimize policies that not only perform tasks effectively but also adhere to specific expectations. tasks.

*One significant challenge* in RL-based fine-tuning of mobile agents is the lack of support for efficient online fine-tuning. Existing fine-tuning approaches over mobile agents primarily rely on static offline datasets like **AitW** (Rawles et al., 2024b) and **AndroidControl** (Li et al., 2024), which fail to capture the dynamic and evolving nature of mobile applications. Frequent updates and new elements like advertisements cause distribution shifts that offline-trained agents struggle to handle, leading to failures in real-world deployments.

*The second challenge* is the need for Reinforcement Learning (RL) algorithms that can operate efficiently within a distributed framework. Asynchronous data collection introduces algorithmic difficulties: non-stationary data distributions hinder convergence, and delays between policy updates and data collection can cause agents to act on outdated policies, degrading performance.

These challenges motivate us to develop DISTRL, as illustrated in Figure 1. DISTRL is a novel and scalable reinforcement learning (RL) fine-tuning pipeline specifically designed for on-device mobile control agents on Android, featuring **Centralized Training** and **Decentralized Data Acquisitions**. Our main contributions are:

**1. Scalable and Asynchronous Data Acquisition Architecture**: DISTRL introduces a decoupled and asynchronous framework that deploys RL agents across heterogeneous worker devices for remote data collection, enhancing training efficiency and scalability (§ 4).

**2. Advanced RL Algorithm for Centralized Training**: We develop A-RIDE, a novel off-policy reinforcement learning algorithm tailored for distributed and asynchronous data utilization, which prioritizes significant experiences to improve sample efficiency while encouraging exploration (§ 5).

In practice, we validate our framework using a T5-based multimodal generation architecture with 1.3B parameters (details in Appendix A.5.1) to efficiently handle both vision and language inputs. To the best of our knowledge, DISTRL *is the first deployable and scalable autonomous RL fine-tuning system for online mobile device control tasks in a distributed environment.*.

## 2 RELATED WORKS

### 2.1 MULTI-MODAL ON-DEVICE CONTROL AGENTS

Recent advancements in pre-trained Large Language Models (LLMs) and Multimodal LLMs (MLLMs) have revolutionized on-device control agents, moving beyond early methods like behavioral cloning or reinforcement learning (Osa et al., 2018; Mnih et al., 2015; Shi et al., 2017). Early agents simulated mouse clicks and typing (Shi et al., 2017; Humphreys et al., 2022) but faced scalability and adaptability challenges.

Modern approaches use pre-trained models with zero or few-shot prompting and fine-tuning for enhanced capabilities. WebGPT (Nakano et al., 2021) employ fine-tuned models for web browsing, while WebAgent (Gur et al., 2023) generates web code using T5. AppAgent (Yang et al., 2023b) and MobileAgent (Wang et al., 2024a) act as drivers, enabling the LLMs to explore and act on mobile device environments. Training multimodal device control agents poses challenges like pixel-level interactions and variability in device ecosystems. Many rely on proprietary Vision-Language Models (VLMs) and wrappers for GUI visual grounding (Driess et al., 2023; Reid et al., 2024), but without fine-tuning, they are limited by the base models (Driess et al., 2023).

Several works fine-tune Vision-Language Models (VLMs) using demonstration data, such as AutoUI and CogAgent (Kapoor et al., 2024; Zhan & Zhang, 2023); however, models trained on static datasets often struggle with the variability of real-world environments (Jiang et al., 2023). Others employ filtered imitation learning with autonomously collected data (Pan et al., 2024; Lai et al., 2024). While DigiRL (Bai et al., 2024) supports on-device reinforcement learning (RL) fine-tuning, it encounters significant inefficiencies in parallel environments. Specifically, DigiRL's multi-machine setup relies on a fully synchronous data acquisition process, causing faster workers to idle while waiting for slower ones. This approach is impractical in real-world scenarios where task durations can vary by up to 100 times, ranging from seconds to over ten minutes.

Our extensive case studies reveal significant limitations in these prior works: advanced MLLMs like GPT-4V (Achiam et al., 2023), SFT-based agents like AutoUI (Zhan & Zhang, 2023), and state-of-the-art mobile control agents like DigiRL (Bai et al., 2024) exhibit numerous failure modes (detailed analysis in Appendix A.1). In particular, DigiRL's lack of efficient distributed learning algorithms severely limits its scalability in dynamic, parallel settings. To address these limitations, we introduce DISTRL, a scalable and asynchronous RL fine-tuning pipeline designed for efficient distributed mobile control agent training.

### 2.2 REINFORCEMENT LEARNING FOR ON-DEVICE AGENT FINE-TUNING

Reinforcement Learning from Human Feedback (RLHF) is widely used to fine-tune LLMs to align with human preferences (Stiennon et al., 2020; Ouyang et al., 2022). In device control tasks, similar approaches use imitation learning from human-labeled evaluations, but RLHF is labor-intensive due to the need for human annotations (Ouyang et al., 2022; Bai et al., 2022). Recent advances in MLLMs (Alayrac et al., 2022; Reed et al., 2022; Li et al., 2023) show impressive multimodal capabilities but often produce incorrect outputs that deviate from human preferences (Ouyang et al., 2022; Ziegler et al., 2019; Bai et al., 2022; Stiennon et al., 2020). Reinforcement Learning from AI Feedback (RLAIF), using AI labelers as proxies, offers an alternative (Yu et al., 2024; Lee et al., 2023). For on-device tasks, AI evaluators assess task completion using prompts and screenshots (Bai et al., 2024; Lee et al., 2023; Yu et al., 2024; Chen et al., 2024).

Previous RL research focused on single-turn tasks, limiting their effectiveness for multi-step problems (Reed et al., 2022; Liang et al., 2023). To address this, we developed a simplified off-policy multi-turn RL algorithm, A-RIDE, which learns from suboptimal online interactions, reducing complexity and accelerating convergence compared to previous value-based methods (Driess et al., 2023; Yao et al., 2023). This approach is effective for large-scale applications like Android device control.

### 2.3 SCALABLE AND DISTRIBUTED RL FRAMEWORK

Scalable reinforcement learning frameworks like Ray RLlib (Liang et al., 2018) enable distributed training by parallelizing policy learning across CPUs and GPUs. RLlib supports various algorithms and efficiently manages neural network training, but it assumes that data collection can be simulated or parallelized within the same infrastructure, limiting their applicability to real-world on-device control tasks involving heterogeneous mobile devices with varying task durations and network conditions.

Moreover, *IMPALA* (Espeholt et al., 2018) and *IMPACT* (Luo et al., 2019) are influential distributed reinforcement learning algorithms, but we are not able to adopt them directly as baselines for our distributed RL system in mobile device control due to their limitations in addressing the unique challenges of this environment. Specifically, these algorithms inadequately handle the fluctuating online experiences inherent in real-world mobile interactions, lack efficient buffer management for distributed cases, offer limited support for off-policy reinforcement learning models, and do not provide the necessary scalability and system optimizations required for distributed mobile control. Implementing them directly would necessitate substantial modifications to manage communication, schedule worker roles, and handle queues and replay buffers effectively. Therefore, we developed our own approach that builds upon the foundational concepts of IMPALA but extends them to meet the specific requirements of on-device control in mobile environments.

## 3 PROBLEM SETUP AND PRELIMINARIES

As presented in Figure 2, we model the on-device control problem as a finite-horizon Markov Decision Process (MDP) $M = \{S, A, T, R, \mu_0, H\}$. Here, $S$ denotes the set of GUI states, represented by screenshots or visual observations of the device screen. $A$ represents the set of actions available to the agent, such as touch events at specific screen coordinates. The state transition function $T : S \times A \times S \to [0, 1]$ defines the probability of transitioning from one state to another given an action. The reward function $R : S \times A \to \mathbb{R}$ provides sparse rewards, typically positive upon task completion. $\mu_0$ is the initial state distribution, and $H$ is the finite horizon of the episode.

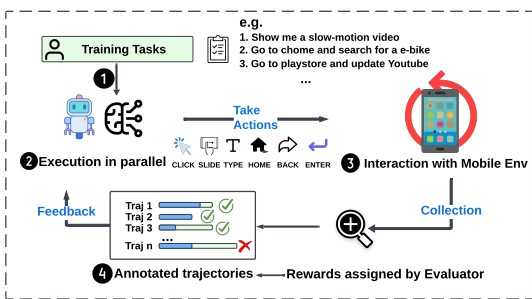

**Figure 2:** Reinforcement Learning dynamics and auto evaluation for fine-tuning the on-device agent.

At each timestep $t$, the mobile agent observes a state $s_t \in S$, selects an action $a_t \in A$ according to its policy $\pi(a_t|s_t)$, receives a reward $r_t = R(s_t, a_t)$, and transitions to the next state $s_{t+1}$. The agent's objective is to maximize the expected cumulative reward $\mathbb{E}_\pi \left[ \sum_{t=0}^{H} r_t \right]$ over the episode. Given the asynchronous nature of distributed data generation in DISTRL, trajectories are collected under behavior policies $\pi_b$ and used to optimize a target policy $\pi$. This setup requires robust off-policy learning algorithms to correct for discrepancies between $\pi_b$ and $\pi$.

A critical component of our RL framework is the ability to obtain reliable reward signals in real-time. To achieve this, we utilize **Gemini-1.5-pro** (Reid et al., 2024) as an autonomous evaluator to assess whether the agent has successfully completed the task at each state. The evaluator receives the current observation, composed of the task description and a screenshot of the device, and outputs a reward signal. Specifically, the evaluator assigns a reward $r_t = 1$ if the screenshot indicates successful task completion and $r_t = 0$ otherwise. Details of how we implemented the auto-evaluation can be found in Appendix A.2.

## 4 SYSTEM DESIGN

DISTRL is an asynchronous distributed reinforcement learning framework for scalable and efficient training of mobile agents. By decoupling trajectory collection from policy learning and doing both in parallel, it leverages distributed working machines for CPU-intense agent-environment interactions and GPU servers for policy training. This separation optimizes efficiency, scalability, and resource utilization by aligning tasks with appropriate hardware. Moreover, such decoupled and asynchronous design offers several key advantages: it improves scalability as data collection scales with more working machines providing the mobile environment, even if there are large performance gaps between them, optimizes resource utilization by assigning tasks to suitable hardware, and it improves policy quality through richer, more diverse datasets from multiple devices, enhancing robustness and generalization capabilities. The details of our system are presented as follows:

As illustrated in Figure 3, DISTRL employs a host-worker architecture consisting of a central **Host Learner** (Left side in Figure 3) and multiple **Workers** (Right side in Figure 3) which can be het-

**Figure 3:** Illustration of the high-level workflow of DISTRL System.

erogeneous devices: i.e. machines of various specifications, running android emulators or being connected with mobile devices, providing the android interaction environments. These components work together to train agents through asynchronous data collection and distributed policy updates.

**Host Learner:** Host Learner orchestrates the policy training process using powerful GPUs. It maintains a **Circular Replay Buffer** (details in Appendix A.4) that stores the trajectories collected from the workers. The training loop processes this data by applying reinforcement learning algorithms to update the policy. To manage incoming data efficiently, a **FIFO Trajectory Queue** receives experiences from the workers and organizes them for training.

The host learner updates the policy using tailored regularization to promote worker exploration and priority-based sampling to efficiently utilize diverse data. Additionally, to maximize the use of the large-scale and diverse data collected, and to avoid excessive learning on redundant or similar data, the learner employs priority-based sampling techniques. Updated policies are then distributed to workers, creating a continuous cycle of experience collection and policy refinement. Detailed algorithmic design is presented in Section 5.

**Workers:** Workers operate in parallel, each managing its own Android environments with Android Emulators or actual Android devices through multi-threading. Each thread in the workers executes the policy received from the host learner and interacts with the environment through an Agent. The agent queries the environment, receives observations, and generates actions based on the current policy. Each worker collects trajectories—sequences of actions, observations, and rewards—during its interaction with the emulator.

To facilitate efficient simulation, workers use **Environment Snapshots**, allowing them to reset the emulator to specific states. The result trajectories from the collecting threads are asynchronously sent back to the host learner to be added to the replay buffer for training. The asynchronous architecture enables diverse worker machines to efficiently collect data independently without interference, maximizing their individual contributions to the system.

On the whole, DISTRL employs asynchronous RL to address the challenges of online RL in dynamic, real-world settings. Each thread in workers operates independently, executing tasks and generating learning trajectories at its own pace, which accommodates variability in task durations and system latencies. Data produced by the working threads is queued and processed by the host learner, which updates the global policy based on the collected trajectories. The updated policy is asynchronously distributed back to the workers, allowing for independent and non-blocking policy updates. Further details are elaborated in Appendix A.3.2.

## 5 METHODOLOGY

In this section, we introduce **A-RIDE**, the core reinforcement learning algorithm employed in DISTRL to fine-tune agents for device control tasks in distributed environments characterized by limited on-device resources, asynchronous data generation, and distributed constraints. Traditional on-policy algorithms like Proximal Policy Optimization (PPO) (Schulman et al., 2017) and Advantage Actor-Critic (A2C) (Mnih, 2016) are inefficient in these settings due to their reliance on synchronous data collection and policy updates, leading to sample inefficiency and delayed learning. Unlike existing off-the-shelf RL models that have been widely applied in various domains (Wang & Yoneki, 2024; Wang & Shi, 2021; Wang et al., 2024b; 2025; Sun & Wang, 2022; Sun et al., 2022), our approach, A-RIDE (**A**dvantage-based **R**etrace **I**mproved by **D**istributed Prioritized **E**xperience Replay), addresses these challenges through two key novel components: **Retrace** (§ 5.3) and **DPER** (§ 5.2), enhancing exploration efficiency, maintaining policy robustness, and improving training efficiency through robust exploratory behavior and prioritization of informative experiences. This

enables DISTRL to achieve stable and efficient learning in real-world device control tasks (see Appendix A.4 for more methodology details).

## 5.1 A-RIDE: THE BACKBONE OF DISTRL

In general, our approach enhances policy gradient updates by extending the Generalized Advantage Estimation (GAE) framework (Schulman et al., 2015) to better suit asynchronous, distributed environments common in device control tasks. Learning from GAE to estimate the advantage function directly, we introduce another stable and smart way to estimate advantages by maintaining two separate networks: one for the trajectory-level value estimation ($V_{\text{traj}}$), which serves as the labeler, and one for the state value function $V(s)$.

**Trajectory-Level Value Estimation** We maintain a trajectory-level value estimator $V_{\text{traj}}$, parameterized by a neural network $\theta$. This estimator serves as a labeler, assigning rewards to trajectories and filtering the replay buffer to retain only high-value trajectories. The trajectory-level value estimator is trained using a maximum likelihood estimation (MLE) loss function:

$$\min_\theta \mathcal{L}(V_{\text{traj}}) = -\mathbb{E}_\nu \left[ r(s_H, a_H) \log V_{\text{traj}}(s_H, a_H) + (1 - r(s_H, a_H)) \log \left(1 - V_{\text{traj}}(s_H, a_H)\right) \right]$$

where $r(s_H, a_H)$ is the reward at the time horizon step $H$, and $\nu$ denotes the distribution over trajectories. This estimator helps identify and retain valuable trajectories for more efficient training.

**State-Value Function Estimation** We also maintain a separate value network $V(s_t; \phi)$, which estimates the state value function $V(s_t)$, representing the expected return from state $s_t$. Instead of directly regressing on scalar returns, we train the value network to predict the probability that the Monte Carlo return $G_t$ is positive, transforming value estimation into a binary classification task.

The value network is trained with the following loss function: $\mathcal{L}(V) = \mathbb{E}[-\mathbb{I}[G_t > 0] \log V(s_t; \phi) - (1 - \mathbb{I}[G_t > 0]) \log(1 - V(s_t; \phi))]$, where $V(s_t; \phi)$ is the predicted probability that $G_t > 0$, $G_t = \sum_{k=t}^{H} \gamma^{k-t} r_k$ is the Monte Carlo return from timestep $t$, and $\mathbb{I}[G_t > 0]$ is an indicator function that equals 1 if $G_t > 0$ and 0 otherwise. The value network parameters $\phi$ are optimized by: $\phi^* = \arg\min_\phi \mathcal{L}(V; \phi)$.

**Advantage Computation** With the trajectory-level rewards and state value estimates obtained, we compute the advantage function $A(s_t, a_t)$ using one-step estimation: $A(s_t, a_t) = Q(s_t, a_t) - V(s_t) = r(s_t, a_t) + \gamma V(s_{t+1}) - V(s_t)$, which correctly represents the advantage function as per the policy gradient theorem. Here, $r(s_t, a_t)$ includes signals of immediate rewards (see § A.4.2). The advantage and value functions are further modified by off-policy corrections, which will be elaborated in Section 5.2.

**Policy Optimization with Robust Regularization** Finally, the policy network optimizes the following loss:

$$\mathcal{L} = -\mathbb{E}_\mu \underbrace{\left[\rho_t A(s_t, a_t) \log \pi(a_t|s_t)\right]}_{\text{Update direction and fine-tuning loss}} - \beta \, \mathbb{E}_\mu \underbrace{\left[\mathbb{H}\left(\pi(a_t|s_t)\right)\right]}_{\text{Regularization term}} + \lambda \, \mathbb{E}_\mu \underbrace{\left[\mathcal{P}_{\text{invalid}}(a_t)\right]}_{\text{Action Penalty}} \quad (1)$$

where $\rho_t = \pi(a_t|s_t)/\mu(a_t|s_t)$ is the importance sampling ratio between the target policy $\pi$ and the behavior policy $\mu$, $\mathbb{H}$ is the entropy term for prevention of overfitting (see § 5.3), $\mathcal{P}_{\text{invalid}}(a_t)$ imposes a penalty on actions deemed invalid based on task-specific criteria, $\beta$ controls the strength of entropy regularization, and $\lambda$ modulates the penalty's influence. The penalty is assigned using validation through pre-trained LLMs like Gemini (Reid et al., 2024)), ensuring that inappropriate actions are penalized. This formulation encourages the agent to explore a diverse set of actions while constraining it to generate valid and meaningful commands, thereby enhancing both exploration and policy robustness when dealing with online non-stationarities.

This formulation enhances our method over traditional GAE and DigiRL (Bai et al., 2024) by: **Incorporating Importance Sampling**: The term $\rho_t$ ensures that the policy updates remain stable even when learning from off-policy data, which is common in asynchronous, distributed environments. **Adding Entropy Regularization**: The entropy term $\mathbb{H}(\pi(a_t|s_t))$ encourages more explorations. **Penalizing Invalid Actions**: The penalty term $\mathcal{P}_{\text{invalid}}(a_t)$ discourages the selection of inappropriate or nonsensical actions. Details of achieving the penalties can be found in Appendix A.4.2.

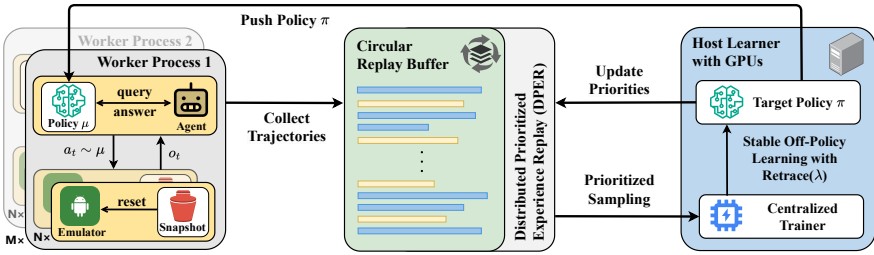

**Figure 4:** Backbone of DISTRL: A-RIDE - Reinforcement Learning-based Fine-Tuning

## 5.2 OFF-POLICY CORRECTION: RETRACE

To enhance the estimation of the state-value function $V(s_t)$ in off-policy and asynchronous settings, we apply Retrace($\lambda$) corrections directly to $V(s_t)$. The Retrace algorithm extends the **TD**($\lambda$) method to off-policy learning by incorporating importance sampling ratios and a trace decay parameter $\lambda$. Specifically, we update $V(s_t)$ using the correction term $\delta_t$, computed as: $V(s_t) \leftarrow V(s_t) + \delta_t$, $\delta_t = \sum_{k=t}^{H} \gamma^{k-t} (\prod_{i=t+1}^{k} c_i)[r_k + \gamma V(s_{k+1}) - V(s_k)]$, where $c_i = \lambda \min(1, \rho_i)$ with $\lambda \in [0, 1]$ being the trace decay parameter, and $\rho_i$ is the importance sampling ratio as mentioned before. This correction term effectively adjusts the value estimates using future rewards and importance sampling, enabling off-policy learning while mitigating variance due to importance weights. By applying Retrace($\lambda$), we improve the estimation of $V(s_t)$ in off-policy settings, enhancing the stability and convergence of the value network.

## 5.3 DISTRIBUTED PRIORITIZED EXPERIENCE REPLAY (DPER)

To improve sample efficiency, we employ **Distributed Prioritized Experience Replay (DPER)**. For each trajectory $\tau = \{(s_t, a_t, r_t, s_{t+1})\}_{t=0}^{H}$, we compute the priority $p(\tau)$ as: $p(\tau) = w_1 \overline{|\delta|} + w_2 \overline{\rho} + w_3 \overline{\mathbb{H}}$, where $\overline{|\delta|}$ is the average absolute temporal-difference (TD) error over the trajectory, calculated as $\delta_t = r_t + \gamma V(s_{t+1}) - V(s_t)$; $\overline{\rho}$ is the average importance sampling ratio $\rho_t$; and $\overline{\mathbb{H}}$ is the average policy entropy, $\mathbb{H}_t = -\log \pi(a_t|s_t)$, encouraging exploration by encouraging policy uncertainty, thus avoiding early convergence to suboptimal policies during training in dynamic environments. The weights $w_1$, $w_2$, and $w_3$ balance the contributions of each component, which is selected by grid-search (See Appendix A.4.3). Trajectories with higher priorities are replayed more frequently, focusing learning on the most informative experiences. Priorities are periodically updated based on the latest policy, recalculating them to focus learning on the most informative experiences, ensuring continual adaptation to evolving behavior policies.

## 6 EXPERIMENTS

To evaluate the performance of DISTRL on challenging Android device control tasks, we conducted extensive experiments. Our primary goal is to determine whether DISTRL can produce agents that effectively learn from autonomous online interaction. We present the experimental environment in Section 6.1, the baseline and benchmarks in Section 6.2, validation of our evaluator in Section 6.3, training performance in Section 6.4, and on-device task evaluations in Section 6.5. Additionally, we present ablation studies on our approach's components in Section 6.6.

### 6.1 EVALUATION ENVIRONMENT

Our evaluation environment consists of a host learner with 4 NVIDIA V100 GPUs for intensive policy training and two worker machines with 8 NVIDIA Tesla T4 GPUs and 96 vCPUs each, supporting parallel emulation. This setup leverages 192 vCPUs to run multiple emulators concurrently, enabling scalable distributed reinforcement learning experiments.

### 6.2 BENCHMARKS AND BASELINE METHODS

To comprehensively validate our approach, we utilize both the *General* and *web shopping* tasks for training and testing. Specifically, our training set is derived from enhanced online task instructions, which are composed of **AitW** (Rawles et al., 2024b), **AndroidWorld** (Rawles et al., 2024a), and expert-curated task sets. We fine-tune our model on this combined training set and evaluate performance on the corresponding test subsets derived from **AitW**. Our analysis focuses on training efficiency using the *General Tasks*, which include fundamental application operations and informa-

tion retrieval tasks. Additionally, we assess the agent's performance on both *General Tasks* and *Web Shopping Tasks* to evaluate its capability in handling domain-specific instructions, addressing the significant task distribution gap. Detailed descriptions of the datasets used are provided in Appendix A.5. Our baseline methods[†] include:

- **DigiRL** (Bai et al., 2024): The state-of-the-art framework prior to our work, which integrates RL fine-tuning with visual language models (VLMs) and provides a reproducible training process. We consider its both single and multi-machine settings in online mode.

- **AutoUI** (Zhan & Zhang, 2023): A simple explorative mobile agent equipped with VLMs under supervised fine-tuning.

- **GPT-4V** (OpenAI et al., 2024) and **Gemini 1.5 Pro** (Reid et al., 2024): Equipped with exploration drives-**AppAgent** (Yang et al., 2023b) to facilitate learning from the environments.

## 6.3 VALIDATION OF OUR EVALUATOR

In our experiments, we used the last screenshot along with the last two actions as input to the VLM Evaluator (Gemini) for prompting. This approach provides a concise yet informative context for the evaluator. As shown in Figure 5, when testing different policy models on *General* subsets from **AiTW**, incorporating additional context, such as longer trajectory information, negatively impacts evaluation accuracy. Specifically, the discrepancy between the evaluator's output and human assessments was less than 2% under our setting. On one hand, our algorithm leverages score assignments based on final steps and states, achieving a balance between computational efficiency and evaluation accuracy, on the other hand providing more context led to performance drops, significantly higher computational costs, and increased usage of evaluator LLMs, thereby straining our budget.

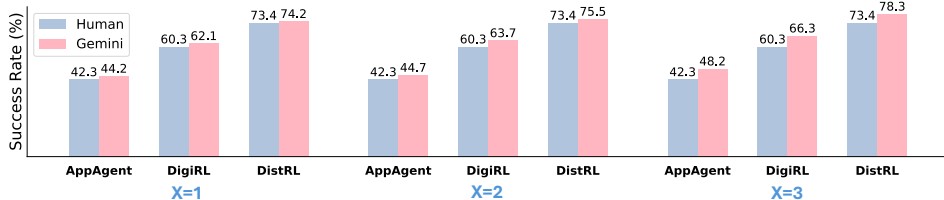

**Figure 5:** Performance correlation between automated and human evaluation across different agents, using the last **X** images (validated on the whole *General* subsets from **AiTW**)

In fact, empirical tests show that large models like Gemini tend to classify tasks as successful by identifying evidence of success in provided screenshots. The more information Gemini receives, the higher the probability it judges the task as successful. In few-shot scenarios with seven or eight images, adding more tokens causes token explosion, leading to hallucinations. These are reasons for our choice to limit context (instead of longer or even entire trajectories).

## 6.4 TRAINING PERFORMANCE

Training efficiency is crucial in reinforcement learning, particularly in complex environments, and is measured by the rate of improvement over time. We compared DISTRL with the existing DigiRL framework. Our results show that DISTRL significantly boosts training efficiency with its distributed, asynchronous design, leveraging multiple machines and GPUs.

In subfigure (a), by 6k seconds, DISTRL achieves a success rate 30% and 40% higher than DigiRL in multi- and single-machine settings, respectively. Even compared with DigiRL that is enhanced with our asynchronous framework (by integrating the DigiRL algorithm into the DISTRL framework to isolate the framework's benefits-named as DigiRL-DistRL Async), DISTRL achieves 10% higher result with faster convergence speed. Subfigure (b) illustrates the proportions of success rates above 60% and 80% in different training phases, representing key stages in the learning curve. DISTRL maintains higher proportions of success rates above these thresholds compared to DigiRL, showing a faster convergence and higher stability. These improvements are attributed to our asynchronous architecture and tailor-made algorithm for efficient data collection and sampling.

Subfigure (c) highlights DISTRL's superior data collection efficiency, accumulating 800 trajectories in 6k seconds, compared to DigiRL's 300 in a multi-machine setting. While subfigure (d)

---

[†]A qualitative comparison among methods and more details can be found in Table 3 in Appendix A.5.1

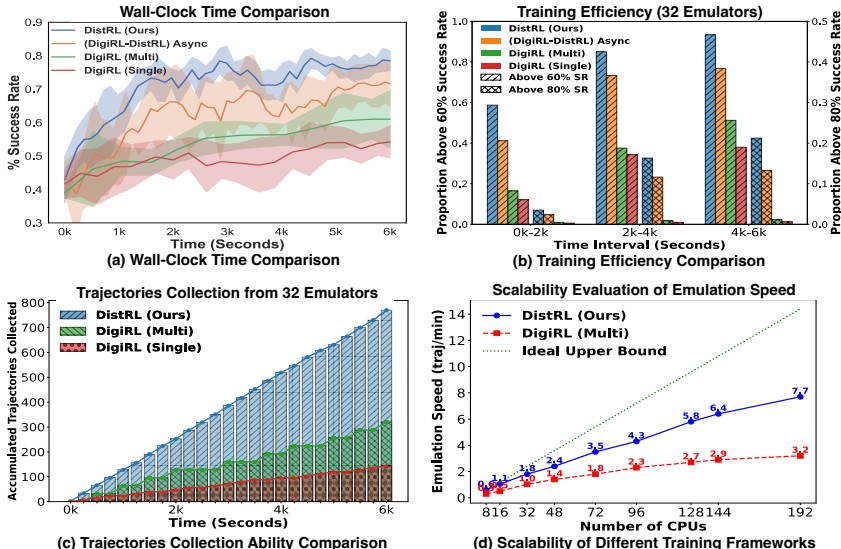

**Figure 6:** Training performance (32 emulators) between the current state-of-the-art method (Di-giRL) and DISTRL, highlighting the enhanced efficiency of DISTRL's distributed framework during online training. **(a)** Wall-clock time comparison **(b)** Training efficiency comparison. **(c)** Accumulated trajectories collection ability comparison. **(d)** Scalability of different training frameworks

demonstrates DISTRL's scalability. It achieves a collection speed of approximately 7.7 trajectories per minute with 192 CPUs, with nearly linear scalability, closely approaching the **Ideal Upper Bound**—perfect linear scalability with no overhead from communication or error handling. This ideal upper bound is determined by assuming each CPU operates independently and continuously with a stable speed, profiled by measuring the collection speed when a single CPU handles the task.

Additionally, Table 1 also presents the final training performance at convergence or after extended training time budgets (will be explained in the subsequent subsection), demonstrating the superior long-term performance of DISTRL compared to the baselines.

## 6.5 PERFORMANCE EVALUATION OF AGENTS TRAINED WITH DISTRL

We evaluate the end-to-end performance of agents trained with DISTRL against other frameworks, including on-device control agents, using subsets of both the **AitW** training and test sets. The primary metric for evaluation is the success rate across *General* and *Web Shopping* tasks. To ensure a fair comparison, we allocate extensive fine-tuning time for DigiRL in single-machine and synchronous multi-machine configurations, typically allowing 2 times the convergence time required by our asynchronous DISTRL multi-machine setup. Despite this generous tuning period, baseline methods often fail to achieve stable performance due to inherent inefficiencies in their synchronous designs, which hinder effective utilization of additional training time.

The results in Table 1 and Figure 7.(a) demonstrate the superior performance of our DISTRL framework over other agents across all evaluated settings. In the *General* test set, DISTRL achieves a success rate of 73.2%, showing a relative improvement of approximately 19.6% over DigiRL (multi) and 22.2% over DigiRL (single). In the *Web Shopping* test set, DISTRL attains a success rate of 68.5%, outperforming DigiRL (multi) by about 14.4% and DigiRL (single) by 14.9%. This significant enhancement is attributed to DISTRL's design for pure asynchronous task collection procedures and its advanced algorithm for efficiently utilizing diverse incoming trajectories, leading to better generalization and higher success rates.

The prompting-based methods, such as AppAgent combined with GPT-4v or Gemini, show considerably lower success rates, not exceeding 45.3% in any test setting. These methods lack adaptive learning capabilities on real-time large-scale interaction data, leading to poorer performance and higher susceptibility to task variability. AutoUI, another learning-based agent fine-tuned by supervised knowledge, also underperforms with success rates below 45%, likely due to less efficient exploration strategies and inadequate handling of diverse user instructions.

| Framework Type | Framework Name | General | | Web Shopping | |
|---|---|---|---|---|---|
| | | Training | Test | Training | Test |
| **Prompting** | AppAgent + GPT-4v | 41.4 | 43.0 | 31.2 | 35.2 |
| | AppAgent + Gemini | 39.1 | 45.3 | 30.5 | 32.0 |
| **Learning** | AutoUI | 38.3 | 40.6 | 42.2 | 44.5 |
| | DigiRL (single,online) | $64.6 \pm 1.5$ | $59.9 \pm 2.1$ | $63.3 \pm 1.5$ | $59.6 \pm 3.1$ |
| | DigiRL (multi) | $67.7 \pm 1.3$ | $61.2 \pm 2.4$ | $64.5 \pm 1.1$ | $59.9 \pm 2.8$ |
| | *DistRL (Ours)* | $\mathbf{75.5 \pm 0.2}$ | $\mathbf{73.2 \pm 1.1}$ | $\mathbf{69.8 \pm 0.5}$ | $\mathbf{68.5 \pm 1.7}$ |

**Table 1:** Main comparisons regarding the **success rate** of different agents across various settings. Each experiment is repeated three times and the mean and standard deviation are reported. Results are evaluated with our autonomous evaluator with the 128 user instructions in the train and test set.

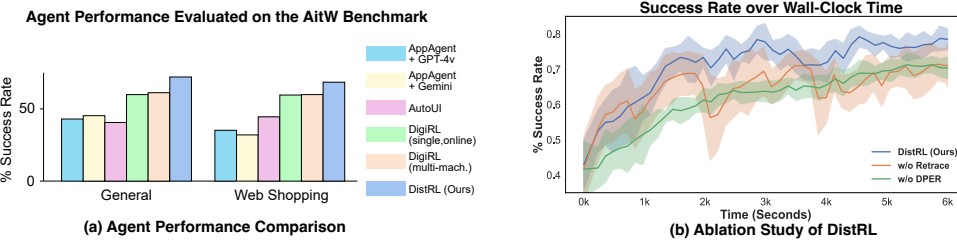

**Figure 7: (a)** Comparison of trained agent performance when evaluated on the **AitW** benchmark.**(b)** Ablation Study of DistRL

## 6.6 ABLATION STUDIES

To understand the contributions of different components in DISTRL, we conduct ablation studies by systematically removing or altering key elements of the algorithm, such as the enhanced Retrace algorithm and Distributed Prioritized Experience Replay (DPER). The results, summarized in Figure 7.(b), demonstrate the significant impact of each component on the task success rate.

**Distributed Prioritized Experience Replay (DPER)** is crucial for accelerating training convergence. Removing DPER results in an **8%** decrease in the success rate, indicating that prioritizing trajectories with higher TD errors and smaller policy discrepancies enables faster and more efficient learning by focusing updates on the most informative experiences. With the entropy term, the prioritization mechanism promotes exploration based on the evolving policy distribution, preventing stagnation during training.

**Retrace Algorithm** is essential for maintaining training stability. Ablating the Retrace algorithm leads to a **6%** drop in success rate and causes sharp decreases in performance during training. This instability arises because Retrace provides off-policy correction, ensuring stable updates even when the agent receives a large number of diverse trajectories.

Overall, the ablation results confirm that both DPER and the Retrace algorithm are integral to the efficiency and robustness of DISTRL.

## 7 CONCLUSION AND FUTURE WORK

In this paper, we introduce DISTRL, an efficient distributed reinforcement learning framework tailored for mobile-based agents tasked with user instructions. Our primary contribution is the development of a robust and scalable pipeline that seamlessly bridges the gap between real-time interactions on mobile devices or emulators and distributed training infrastructures, ensuring efficient and adaptive learning. For future work, we aim to extend the generalization capabilities of DISTRL to a broader range of tasks, focusing on enhancing both the training pipeline and the underlying algorithmic architecture. Additionally, we envision evolving DISTRL into a core backbone for integrating many more Multimodal Large Language Models (MLLMs), allowing for a wider range of applications and evaluations on diverse benchmarks.

ACKNOWLEDGEMENTS

This work was supported by the National Natural Science Foundation of China (Grant Nos. 62422605, 92370132), AWS cooperative funding supports and POWERSENSE TECHNOLOGY LIMITED.

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

## A APPENDIX

### A.1 CASE STUDY

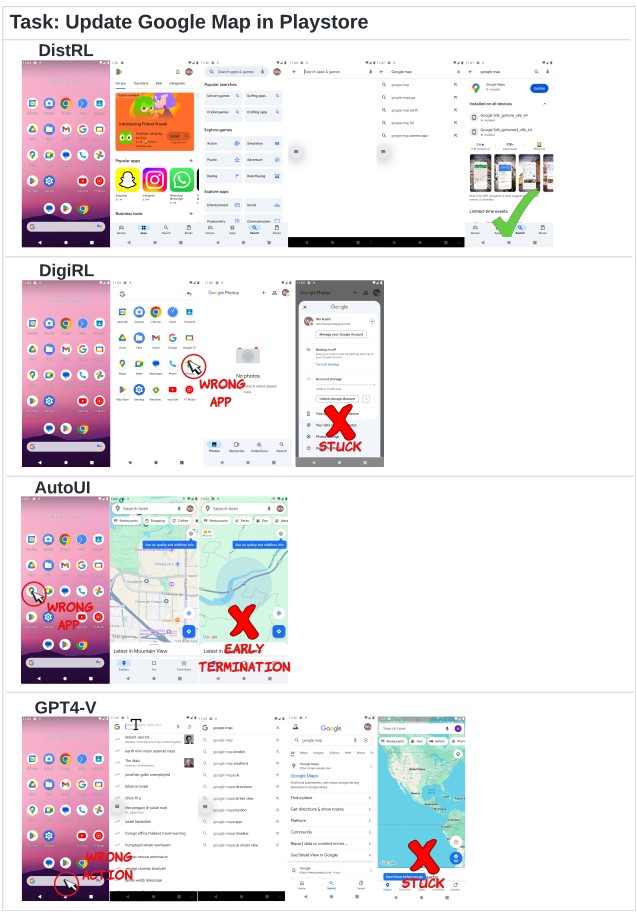

**Figure 8:** Case study on general app operation tasks.

Figure 8 illustrates a type of common case where baseline methods always fails, highlighting the challenges for these device-control agent in real-time app operation tasks.

DigiRL, which was trained on older versions of the system, fails due to discrepancies between its learned knowledge and the current environment. This mismatch in training data leads to a significant error: DigiRL mistakenly opens Google Photos instead of the Play Store. Since the two apps share similar icon features. After opening the wrong app, DigiRL continues to operate within the incorrect environment, ultimately getting stuck in the settings menu of Google Photos. This reveals a significant limitation of offline-training-only agent in adapting to updated environments, especially when visual similarities between app icons lead to misclassification. AutoUI shares a similar issue where it struggles to correctly identify the target application. In this case, it opens Google Maps directly instead of navigating through the Play Store. Its lack of adaptability to new tasks or novel instructions results in failure.

The **AppAgent** with **GPT-4V** takes an alternate route by resorting to web searching, which diverges from the intended method of updating the app. Eventually, this leads to the agent becoming stuck within the Google Maps application itself, indicating that while **GPT-4V** was able to explore different avenues to achieve the goal, it did not follow the expected approach due to the lack of app-specific knowledge.

While the DISTRL, which was actively trained on the real-time newly-updated environment through online-training, could conduct the intended operations accurately and successfully.

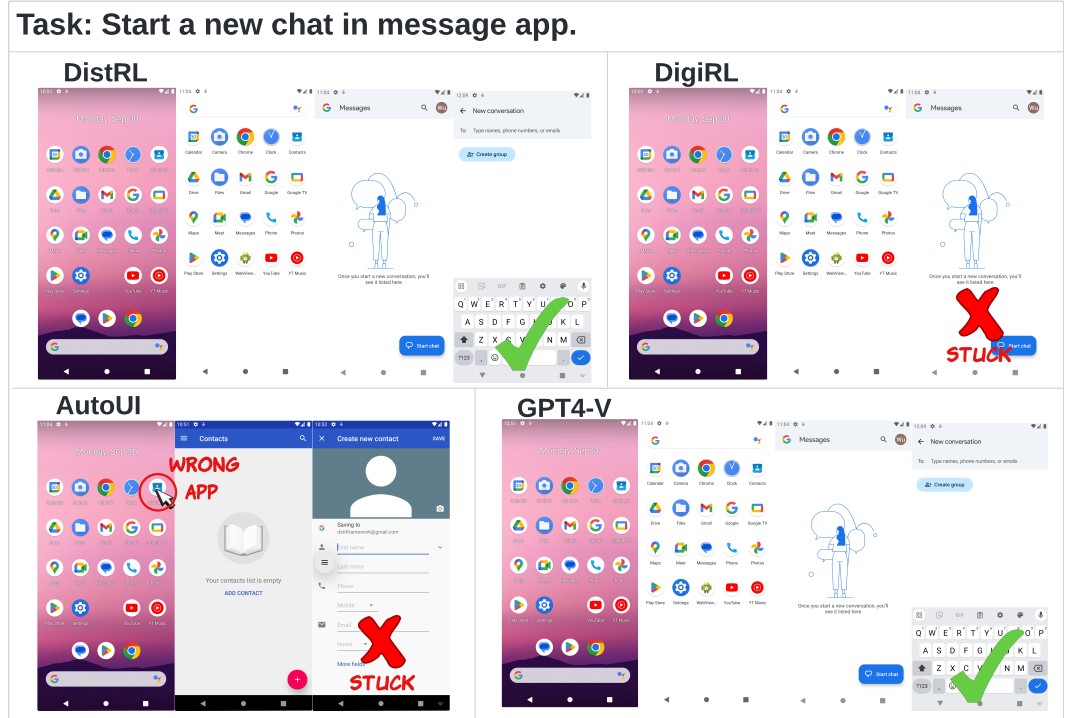

**Figure 9:** Case study on general app operation tasks.

The case study in Figure 9 further illustrates the comparative performance on a simpler general app operation task — starting a new chat in the messaging app.

DISTRL successfully completes the task by efficiently navigating through the app's interface. It opens the correct messaging app and enters the "New Conversation" screen without getting stuck.

In contrast, DigiRL manages to open the correct messaging app but fails to proceed, getting stuck when attempting to start the new chat. This is due to DigiRL's reliance on outdated training data, as it was trained on an older version of the app's interface. In the outdated UI, the intended action (starting a chat) involved interacting with elements in a different layout, and DigiRL cannot adapt to the updated version. This demonstrates the pitfalls of relying primarily on offline data for training without sufficient online fine-tuning to adapt to new UI changes, as seen in modern apps that frequently update their designs.

AutoUI, on the other hand, fails immediately by selecting the wrong app. It opens the Contacts app instead of the messaging app, leading to a failure in completing the task from the very beginning. This reflects a limitation in AutoUI's task understanding and its inability to differentiate between similar apps, further highlighting the weakness of frameworks that lack a robust decision-making process or real-time adaptability.

**GPT-4V**, though not specifically trained for app-specific tasks, performs well in this scenario due to its generalization capability. It opens the correct messaging app and navigates to the "New Conversation" screen successfully. **GPT-4V** is more flexible and suitable for simpler, general-purpose tasks. However, this general-purpose approach may not scale well for more complex tasks where app-specific expertise and interaction nuances are required.

The real cases in the figures emphasizes the critical importance of efficient real-time online learning.

Figure 10 shows a case study on the web shopping task. DISTRL demonstrates relatively smooth and fluid operations, progressing through the steps without hesitation. While it ultimately encounters early termination due to reaching the step limit (horizon), it performs each step with clear transitions and effectively navigates through the sequence. DISTRL shows strong task comprehension and adaptation throughout, but its misunderstanding on the task requirement prevents it from fully com-

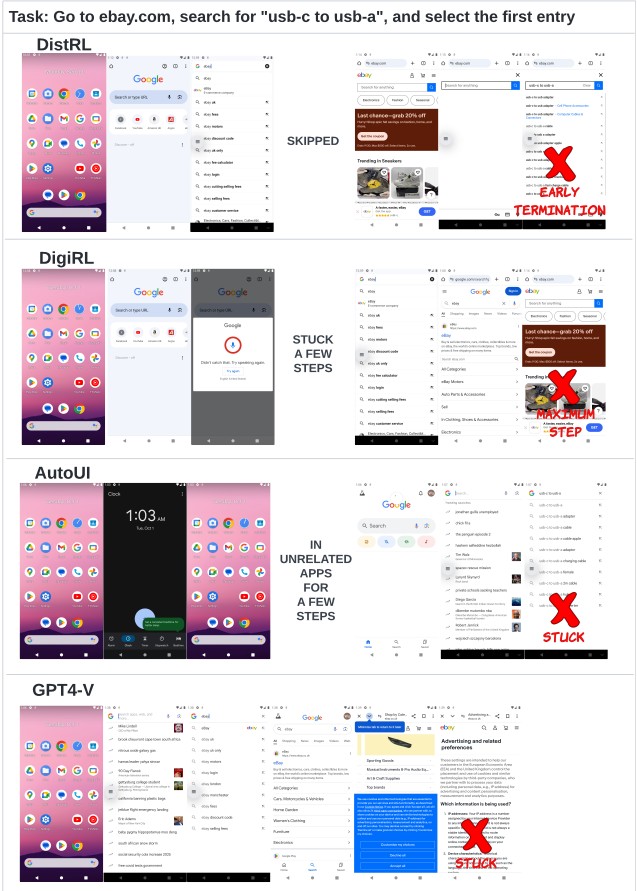

**Figure 10:** Case study on web shopping tasks.

pleting the task. This behavior emphasizes the efficiency of DISTRL's operations and its capacity to generalize across unseen web shopping tasks, even though the task is terminated early.

In contrast, DigiRL faces several challenges during the task. It frequently steps back and forth between pages, struggling with the microphone input. These actions result in unnecessary delays and inefficiencies, which eventually lead it to reach its step budget without successfully completing the task. The back-and-forth behavior indicates a lack of robust policy adaptation, which causes it to get stuck in a loop, unable to make meaningful progress.

AutoUI, on the other hand, wanders into unrelated apps before eventually returning to the task. The lack of focus results in it spending multiple steps outside the task's scope, which ultimately contributes to its failure. This signifies weaknesses in both task planning and execution, as it struggles with distractions and incorrect app selections.

**GPT-4V** follows a similarly smooth approach as DISTRL, but it becomes stuck after selecting a wrong entry into eBay, which triggers the cookie settings of the website. Although **GPT-4V** successfully navigates through several steps, it ultimately fails to get around the emergent pop-up, highlighting its limitation in handling web-specific tasks that require precision and app-specific understanding.

In summary, while DISTRL and **GPT-4V** demonstrate smoother task execution, only DISTRL manages to maintain a consistently structured progression, even though it faces early termination. Meanwhile, DigiRL struggles significantly, exhibiting inefficient operations that lead to step budget exhaustion without meaningful progress. This case study emphasizes the importance of the ability to adapt policies in dynamic environments to complete tasks successfully within budgeted steps.

## A.2 Auto Reward Labeling

### A.2.1 Evaluation with Auto-Evaluator

To generalize the evaluator across a wide range of tasks without manual rule definitions, we leverage a pre-trained LLM with appropriate prompting. The prompt is designed to instruct the LLM to act as an expert evaluator, i.e., **Gemini-1.5-Pro** (Reid et al., 2024), in our practice. An example of such a prompt is provided below:

```
You're an expert in evaluating whether the Screenshot successfully completes the Task.
=====Examples=====
Screenshot: {train_1.png}
Task: Send a message to Evelyn.
Q: What should I expect to see on the screenshot if I've sent a message to Evelyn?
A: I should expect to see an open messaging app with a conversation window showing
a message sent to "Evelyn." The screenshot, however, shows the messaging app's contact list,
but no message has been sent.
Status: failure
```

In this prompt, the evaluator compares the expected outcome of the task with the actual screenshot. By analyzing the visual content and reasoning about the task, the LLM determines task completion.

### A.2.2 Reward Penalty

To capture long-term dependencies in the device control setting, Monte-Carlo (MC) rollouts were employed to compute cumulative returns, which are then propagated backward to inform updates across each transition. However, during the experiments, we observed frequent repeated nonsense actions even in successful trajectories when doing roll-out with AutoUI agent, which sometimes causes unstable convergence during the asynchronous online learning. Thus, we further deployed a reward penalty on unexpected behaviors: accumulative penalty on repetitions and hard penalty on invalid actions.

## A.3 System Design

### A.3.1 Detailed System Description

DISTRL is a distributed reinforcement learning framework designed for scalable, efficient training of mobile agents. It decouples trajectory collection from policy learning, utilizing working machines for agent-environment interactions and GPU servers for policy training. The working devices handle inference and CPU-intense data collection, transmitting data asynchronously to GPU servers for training large language models (LLMs). This separation optimizes efficiency, scalability, and resource utilization by aligning tasks with appropriate hardware.

This decoupled design offers several key advantages. First, it enhances efficiency by preventing resource contention: mobile devices focus on interaction tasks without being slowed by training, and GPUs dedicated to training perform updates without interruption. Notably, different types of GPUs are used; lightweight GPUs or CPUs handle inference and data collection, while high-performance GPUs are employed for intensive training computations. Second, scalability improves as more mobile devices are added, allowing data collection to scale naturally without single-machine hardware limitations. Third, resource utilization is optimized by aligning tasks with suitable hardware, maximizing performance. Dedicated training resources achieve faster convergence by efficiently processing larger batches and complex models. Cost efficiency is enhanced by leveraging existing devices for data collection and appropriately allocating GPU resources based on task requirements, reducing unnecessary hardware investments. Finally, the quality of learned policies improves due to the richer and more diverse dataset collected from multiple devices, enhancing robustness and generalization capabilities.

### A.3.2 Communication between Host Learner and Workers

In our DISTRL framework, communication between the Host Learner and Workers is crucial for synchronizing policy updates and collecting trajectories. We have opted to use SCP (Secure Copy Protocol) over SSH to transfer LoRA weights between the Host Learner and Workers. This choice is based on several practical considerations related to bandwidth, overhead, and deployment flexibility.

Within the AWS environment, network bandwidth between instances can exceed 500 Mbps. As illustrated in Table 2, transferring the LoRA weights (approximately 100 MB) using SCP takes less than 2 seconds on average. This communication overhead is negligible compared to the time required for trajectory collection and policy training.

**Table 2:** Communication Time for Transferring LoRA Weights via SCP

| Network Bandwidth | LoRA Weight Size | Transfer Time |
|---|---|---|
| 500 Mbps | 100 MB | 1.6 seconds |
| 1 Gbps | 100 MB | 0.8 seconds |

Each Worker thread completes approximately 6–10 trajectories per minute. Concurrently, training the policy on the Host Learner, even utilizing 4 V100 GPUs, takes around 120 seconds to perform a single model update. Consequently, the communication time of a few seconds for transferring weights is significantly lower than both the data collection and training durations, introducing no noticeable bottlenecks.

Alternative high-performance communication options like InfiniBand (IB) or RDMA over Converged Ethernet (RoCE) were considered. However, these technologies require specialized hardware and configurations, which are not always available or practical—especially when Workers (mobile devices or emulators) are dispersed across different physical locations or data centers. SCP over SSH offers a flexible and widely supported solution that operates effectively across diverse environments.

In summary, the minimal communication overhead introduced by using SCP over SSH does not adversely affect the overall performance of the DISTRL framework. The simplicity, reliability, and broad compatibility of this approach make it a reasonable and efficient choice for our distributed reinforcement learning system.

## A.4 METHODOLOGY DETAILS

### A.4.1 LIMITATIONS OF TRADITIONAL METHODS

On-policy algorithms such as Proximal Policy Optimization (PPO (Schulman et al., 2017)) and Advantage Actor-Critic (A2C (Mnih, 2016)) require synchronous data collection and updates, leading to inefficiencies in distributed and large-scale environments due to low sample efficiency and synchronization delays.

Standard off-policy methods like V-trace have been popular in distributed RL frameworks (e.g., *IMPALA* (Espeholt et al., 2018)) but can be suboptimal when the divergence between the behavior policy $\mu$ and the target policy $\pi$ is either too small or too large due to clipping mechanisms.

These weights were empirically validated to provide a robust trade-off between bias and variance, enhancing the overall learning efficiency and stability of the reinforcement learning agent in our distributed, asynchronous setting.

### A.4.2 IMPLEMENTATION DETAILS

**Circular Replay Buffer** We utilize a **Circular Replay Buffer** with fixed capacity $N$ to store experience tuples $(s_t, a_t, r_t, s_{t+1}, a_{t+1})$. When the buffer is full, new experiences overwrite the oldest ones, ensuring that the buffer contains the most recent experiences, which is effective in non-stationary environments.

The buffer index $i$ is updated as:

$$i \leftarrow (i + 1) \bmod N. \tag{2}$$

**Enhanced Retrace Algorithm** Retrace($\lambda$) adjusts the importance sampling corrections based on policy divergence. When policies are similar, it fully exploits trajectories through $\lambda$-returns. When they differ significantly, it truncates importance sampling ratios to control variance, ensuring stable and unbiased updates.

**Temporal-Difference Error Calculation** The temporal-difference (TD) error for each step is calculated as:

$$\delta_t = r_t + \gamma V(s_{t+1}) - V(s_t), \tag{3}$$

which represents the discrepancy between predicted and actual rewards, guiding the learning updates.

**Priority-Based Sampling in DPER** In **Distributed Prioritized Experience Replay (DPER)**, trajectories are sampled based on their computed priority to focus on the most informative experiences. The probability of sampling a trajectory $\tau$ is proportional to its priority $p(\tau)$, calculated as:

$$P(\tau) = \frac{p(\tau)^\alpha}{\sum_i p(\tau_i)^\alpha},$$

where $\alpha = 0.5$ controls the extent to which prioritization is applied. A value of $\alpha = 0.5$ provides a balance between uniform sampling (when $\alpha = 0$) and full prioritization (when $\alpha = 1$), allowing the model to benefit from both the prioritization of informative trajectories and a degree of randomness. This balance ensures that less prioritized but potentially useful experiences still have an opportunity to be replayed, helping to prevent overfitting to a narrow subset of the replay buffer.

DPER improves sample efficiency by prioritizing high-value transitions, such as those with significant temporal-difference (TD) errors or high importance sampling ratios. By focusing on these informative transitions, the agent learns more effectively from previously explored states, reducing unnecessary exploration of less relevant areas. Additionally, by maintaining policy entropy, DPER balances exploration and exploitation, accelerating convergence with fewer samples.

**Policy Update Mechanism** The actor (policy network) is updated using gradients derived from the advantage estimates:

$$\nabla_\theta \mathcal{L} = -\mathbb{E}_\mu \left[ \rho_t A(s_t, a_t) \nabla_\theta \log \pi_\theta(a_t|s_t) \right] - \beta \nabla_\theta \mathbb{E}_\mu \left[ \log \pi_\theta(a_t|s_t) \right] + \lambda \mathbb{E}_\mu \left[ \mathcal{P}_{\text{invalid}}(a_t) \right], , \quad (4)$$

where $\theta$ represents the parameters of the policy network, $\rho_t$ corrects for off-policy data, $\mathbb{H}(\pi)$ encourages exploration, and $\mathcal{P}_{\text{invalid}}(a_t)$ penalizes invalid actions. In this formula, the invalid action penalty term $\mathcal{P}_{\text{invalid}}$ penalizes actions that the agent cannot execute, addressing the challenge of entropy regularization, which encourages exploration but can lead to invalid actions. In mobile device control tasks, valid actions such as clicking and typing, while invalid commands, such as "rotate screen" in an unsupported context or interacting with non-existent UI elements, waste resources and hinder learning.

We use **Gemini-1.5-Pro** to evaluate each action $a_t$. If an action is invalid, $\mathcal{P}_{\text{invalid}} = 1$; otherwise, it is 0. The penalty term is integrated into the loss as $\mathcal{L}_{\text{penalty}} = \lambda \cdot \mathbb{E}[\mathcal{P}_{\text{invalid}}]$, where $\lambda$ controls its impact. This approach ensures exploration remains within valid bounds, enhancing learning efficiency and robustness .

**How is $A(s_t, a_t)$ calculated?** The advantage function $A(s_t, a_t)$ is calculated using a one-step advantage estimation approach, as follows:

$$A(s_t, a_t) = r(s_t, a_t) + \gamma V(s_{t+1}) - V(s_t)$$

Here, $r(s_t, a_t)$ comprises two components: the Monte Carlo return from timestep $t + 1$ to the terminal state $s_H$, which is determined by success or failure signals, and hard-coded penalties for repeated actions and certain violations, serving as immediate reward signals. This combination provides a long-term cumulative reward estimate. $V(s_t)$ represents the estimated state value at timestep $t$, $V(s_{t+1})$ is the estimated state value at the next timestep $t + 1$, and $\gamma$ is the discount factor.

**The trajectory value estimation function $V_{\text{traj}}$** The trajectory value function $V_{\text{traj}}$ estimates the expected return from terminal state $s_H$ in sparse/delayed reward environments. It serves as a baseline by labeling reward signals in collected trajectories (analogous to RLHF reward models) and filters/prioritizes trajectories for training. Through this filtering mechanism, $V_{\text{traj}}$ maintains high-quality training data, leading to more effective policy learning.

### A.4.3 HYPERPARAMETER TUNING FOR DISTRIBUTED PRIORITIZED EXPERIENCE REPLAY

To enhance sample efficiency in our **Distributed Prioritized Experience Replay (DPER)** framework, it is crucial to appropriately balance the contributions of the average temporal-difference (TD) error ($\overline{|\delta|}$), the average importance sampling ratio ($\overline{\rho}$), and the average policy entropy ($\overline{H}$). These components inherently operate on different scales, necessitating careful normalization to ensure that no single component disproportionately influences the priority calculation.

Each component contributing to the priority score is normalized to a common scale based on their statistical properties observed during preliminary training runs. The normalization process is as follows:

- **Average Absolute TD Error** ($\overline{|\delta|}$): Normalized by dividing by the maximum absolute TD error observed across all trajectories in the training set. This scaling ensures that $\overline{|\delta|}$ ranges between 0 and 1.

- **Average Importance Sampling Ratio** ($\overline{\rho}$): As importance sampling ratios naturally fall within the range [0, 1], no additional scaling is required.

- **Average Policy Entropy** ($\overline{H}$): Normalized by dividing by the maximum observed entropy value during training, ensuring that $\overline{H}$ also ranges between 0 and 1.

This normalization facilitates a balanced contribution from each component when computing the overall priority, preventing any single factor from dominating the priority score.

The weights $w_1$, $w_2$, and $w_3$ are critical in determining the influence of each normalized component on the priority calculation. To identify the optimal values for these weights, we employed a grid search strategy on a validation set, exploring the following empirical ranges: $w_1 \in \{0.01, 0.1, 0.5, 1.0, 2.0, 5.0, 10.0\}$, $w_2 \in \{0.01, 0.10.3, 0.5, 0.7, 1.0\}$, $w_3 \in \{0.01, 0.1, 0.3, 0.5, 0.7, 1.0\}$.

These ranges were selected based on insights from prior research (Schaul, 2015; Horgan et al., 2018) and preliminary experiments that indicated effective performance within these intervals.

The chosen weights effectively balance the three components, ensuring that:

- **Learning from High-TD Error Trajectories**: By assigning a higher weight to $\overline{|\delta|}$, the framework emphasizes replaying experiences where the model's predictions were significantly off, facilitating targeted learning and faster convergence.

- **Maintaining Exploration**: The weight on policy entropy ensures that the agent continues to explore diverse actions, preventing premature convergence to suboptimal policies.

- **Correcting for Distributional Shifts**: The importance sampling ratio weight allows the algorithm to adjust for changes in the policy distribution, maintaining unbiased updates despite using prioritized replay.

## A.5 EXPERIMENTAL DETAILS

### A.5.1 BASELINE METHODS

We evaluate proprietary vision-language models (VLMs), **GPT-4V**(OpenAI et al., 2024) and **Gemini 1.5 Pro**(Reid et al., 2024), using the AppAgent framework. By applying the prompt from (Yang et al., 2023b), we enable these models to interact effectively with the environment. We assessed the **AppAgent** (Yang et al., 2023b) in an augmented prompting setting, where the agent explores the environment and gathers experience ahead of the inference phase. This collected experience is appended to the test-time prompt, enhancing the model's decision-making capabilities. Unlike learning-based approaches, these methods rely on advanced prompting strategies to accomplish tasks without extensive training.

In our system's implementation, we maintained a consistent and conflict-free approach by exclusively utilizing the T5-based Multimodal Large Language Model (MLLM) architecture (not vanilla pretrained T5, T5-Base normally has 220M, Our Agent (AutoUI-driven) has 1.3B). This encoder-decoder framework facilitates the seamless integration of various agents, enhancing the model's versatility and performance, which is aligned with AutoUI (Zhan & Zhang, 2023) design.

Our T5-based MLLM architecture is inherently designed to support diverse decoder initializations, including the integration of pretrained decoder models. While previous works like AutoUI (Zhan & Zhang, 2023) have utilized their own model weights, we could not directly use AutoUI's model weights to initialize our models because they were trained with AiTW knowledge. Incorporating these weights could introduce unintended biases specific to their tasks as well as prior knowledge to our training and testing data set, which might reduce the credibility and confidence of our system.

Meanwhile, our empirical analysis showed that decoder layers require more diverse initialization patterns than encoder layers to achieve optimal performance in UI-specific tasks, GPT-2 help more than the pre-trained T5 weights. Therefore, the best approach was to manually select a pretrained decoder model to initialize the weights of our T5-based decoder. We chose to initialize the decoder with GPT-2 weights, as this allowed us to leverage GPT-2's pre-trained language generation capabilities, providing a foundational understanding of language that we could further refine through fine-tuning tailored to our specific tasks.

We did not specify the decoder type/weights in the original explanation because we found that fine-tuning efficiency was already satisfactory with GPT-2. We acknowledge that there might be better options available; however, we adopted this approach simply by drawing inspiration from our baseline DigiRL (Bai et al., 2024), which also used GPT-2 to initialize their decoders (This is a trick here). By following a structured and methodical approach from existing baselines, we ensured architectural compatibility and effective integration of the GPT-2 weights into our T5-based decoder.

In Table 3, we compare DISTRL with other frameworks based on scalability, task diversity, and training efficiency.

**Table 3:** Comparison among on-device agents' frameworks based on scalability, task diversity, and training efficiency.

|  | DISTRL (Ours) | DigiRL | AutoUI | AppAgent+MLLMs |
|---|---|---|---|---|
| **Type** | Async. | Sync. | N/A | N/A |
| **Scalability** | ++ | + | N/A | N/A |
| **Task Diversity** | General | Limited | Limited | General |
| **Training Eff.** | High | Low | Low | N/A |
| **Multi-GPUs Sup.** | ✓ | ✓✗ (offline only) | ✗ | ✗ |

### A.5.2 TRAINING AND TEST DATA

The dataset used in this work is based on the Android in the Wild (**AitW**) (Rawles et al., 2024b) and **AndroidWorld** (Rawles et al., 2024a) task set, with enhancements for practical use in fine-tuning agents to control mobile devices and interact with real-world applications. We trained two separate models using two distinct subsets for *General Tasks* and *Web Shopping Tasks*. Each model was trained on its corresponding training subset and evaluated on the respective test subset drawn from **AitW**.

For training, we utilized the *General Tasks* subset from **AitW**, augmented with tasks selected from **AndroidWorld** and several expert curated ones, which includes tasks that require basic to complex application usage and information retrieval. For *Web Shopping Tasks*, we used the subset from **AitW** directly. Each training set consists of more than 400 tasks, allowing the agent to learn from a broad range of apps and websites operations and promote robust learning without overfitting to specific task types.

To avoid cold start issues in our asynchronous reinforcement learning framework, we constructed a warmup trajectory dataset for each task type, each consists of 128 trajectories collected with an initial version of the AutoUI agent. These sets will be fed into the replay buffer at the beginning of the online training.

During testing, we evaluated the models on their respective test sets: 128 tasks for *General Tasks* and 128 tasks for *Web Shopping Tasks*, both sourced from **AitW**. This approach ensures that each model is assessed on the task domain it was trained on, addressing the task distribution gap between general user instructions and domain-specific web shopping instructions.

**General Tasks** The *General Tasks* subset consists of tasks that involve basic application operations and information retrieval. Examples include searching for the latest news, retrieving information about locations, and interacting with mobile apps. To force the agent to operate more on the various applications instead of searching everything through web, we augmented the task set for training with several instructions from **AndroidWorld** and some expert curated tasks, which are typically more complex and application-specific. The training set contains 600 tasks, and the test set includes 128 tasks from **AitW**, facilitating a robust evaluation of *General Tasks* performance. Each task

allows a maximum of 15 steps to complete. Example tasks from the *General Tasks* subset are shown in Table 4.

| Task Set | Task Example |
|---|---|
| AitW | What is the capital of Norway?
Play some music on YouTube. |
| AndroidWorld | Run the stopwatch.
Create a new contact for Jack. Their number is 0123456789. |
| Expert Curated | Check today's events in the calendar.
Check if there is any app to update in Playstore. |

**Table 4:** Examples of task descriptions in the General Tasks subset.

**Web Shopping Tasks**   The *Web Shopping Tasks* subset includes tasks that simulate real-world shopping activities such as searching for products, navigating e-commerce websites, and interacting with shopping carts. Task complexity ranges from simple web navigation to multi-step operations involving product searching and browsing. The training set consists of 500 tasks, and the test set includes 128 tasks from **AitW**, enabling the evaluation of the agent's ability to handle domain-specific instructions. Each task permits up to 20 steps to complete. Example tasks from the *Web Shopping Tasks* subset are presented in Table 5.

| Difficulty | Task Example |
|---|---|
| 1 | Go to ebay.com |
| 1 | Go to costco.com |
| 2 | Go to ebay.com, search for "asus zenbook" |
| 2 | Go to walmart.com, search for "corsair k70" |
| 3 | Go to bestbuy.com, search for "dell xps", and select the first entry |
| 3 | Go to newegg.com, search for "bose soundlink mini", and select the first entry |

**Table 5:** Examples of task descriptions in the Web Shopping Tasks subset.

### A.5.3   Detailed Performance Comparison

Other methods struggle to achieve comparable performance. DigiRL, both in single and multi-machine settings, suffers from inefficiencies in data collection and utilization. The multi-machine version requires extensive collection time due to its low efficiency, hindering its ability to train effectively on diverse tasks, while the single-machine version struggles with scalability issues. These inefficiencies lead to higher variance in performance, as evidenced by the higher standard deviations (up to $\pm 3.1\%$) compared to DISTRL.

Overall, the low variance and high success rates of DISTRL demonstrate its robustness and effectiveness in generalizing across different tasks, emphasizing the advantages of our distributed reinforcement learning approach over existing methods, especially in large-scale, asynchronous settings.

### A.6   Additional Quantitative Experiments

### A.6.1   Failure Modes Analysis

Figure 11 presents a comparative analysis of failure rates across different approaches on the **AitW** *General* and *Web Shopping* subsets. Among the evaluated frameworks, DISTRL consistently exhibits the lowest failure rates across all failure categories, notably excelling in recovering from mistakes and achieving the correct goal.

For the *General* subset, DISTRL demonstrates exceptional performance with failure rates as low as 4% in recovering from mistakes, 12% in getting stuck midway, and 2% in arriving at an incorrect goal. These rates are at least three times lower than those observed in alternative approaches such as AutoUI and DigiRL. This significant reduction in failure rates can be attributed to DISTRL's robust asynchronous distributed reinforcement learning (RL) framework, which facilitates more nuanced and adaptive policy learning. The distributed nature of DISTRL allows for parallel exploration and exploitation of the state-action space, leading to a more comprehensive understanding of task dynamics and improved decision-making accuracy.

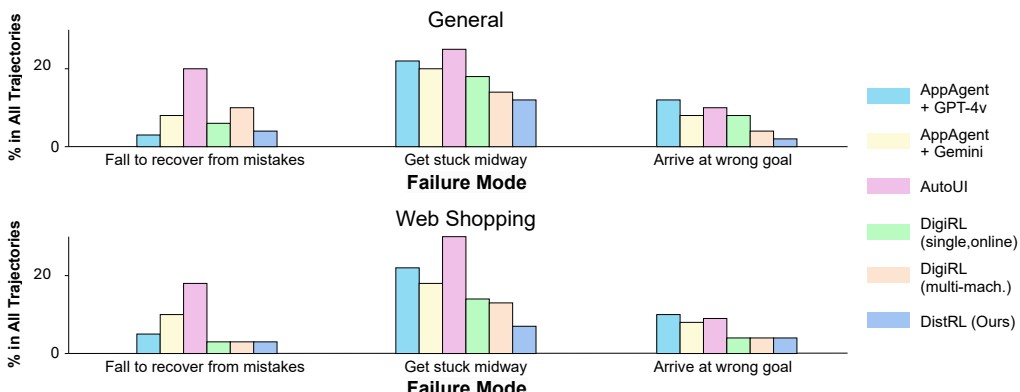

**Figure 11:** Comparison of **failure modes** across different frameworks on the AitW General and Web Shopping subsets.

Similarly, on the *Web Shopping* subset, DISTRL maintains low failure rates of 3% in recovering from mistakes, 7% in encountering mid-task obstacles, and 4% in goal misalignment. These figures represent at least a twofold improvement over competing frameworks, highlighting DISTRL's superior capability in managing complex and dynamic task environments. The ability to effectively handle task complexities is further reinforced by the asynchronous updates in DISTRL, which mitigate issues such as delayed feedback and non-stationary environments that often plague distributed learning systems.

In contrast, frameworks like AutoUI and DigiRL exhibit higher failure rates, which may stem from their less sophisticated policy learning mechanisms or limited scalability in distributed settings. These higher failure rates suggest that these approaches may struggle with tasks that involve intricate dependencies or require rapid adaptation to changing conditions. The limitations observed in these frameworks underscore the importance of advanced distributed learning architectures in developing resilient and efficient agents capable of navigating complex, real-world environments.

Overall, the superior performance of DISTRL across multiple failure modes underscores its effectiveness in building robust agents. This robustness is crucial for applications where reliability and precision are paramount, such as automated web interactions and general task execution. Future work may explore further enhancements to the distributed framework, such as incorporating more sophisticated exploration strategies or leveraging transfer learning to extend capabilities to even more diverse task domains.

A.6.2 GENERALIZATION PERFORMANCE ON AITW SUBSETS

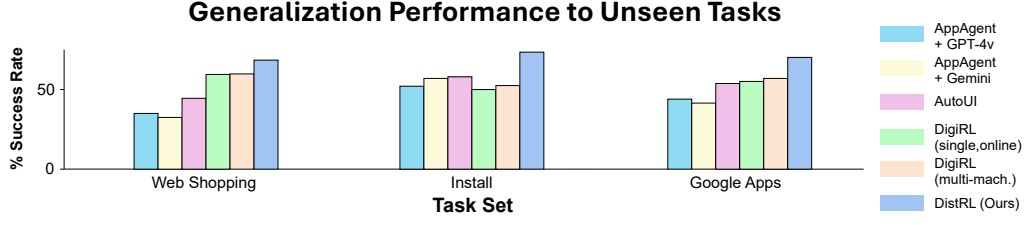

**Figure 12:** Generalization performance across different AitW subsets. The agents were trained on the General task set and evaluated on 128 tasks per subset.

Figure 12 illustrates the generalization performance of various frameworks across the *Web Shopping*, *Install*, and *Google Apps* subsets of the **AitW** dataset. The agents were trained exclusively on the *General* task set, and their ability to generalize was assessed on the first 128 tasks within each respective subset.

DISTRL consistently outperforms its counterparts, achieving accuracies of 68.5% on *Web Shopping*, 73.5% on *Install*, and 70.2% on *Google Apps*. These results highlight DISTRL's superior generalization capabilities, which can be largely attributed to its robust distributed learning approach. The asynchronous distributed RL framework employed by DISTRL enables the agent to learn from a diverse set of experiences concurrently, fostering a more versatile and adaptable policy that can transfer effectively across different task domains.

In contrast, frameworks such as DigiRL and **AppAgent** exhibit markedly lower generalization performance. DigiRL and **AppAgent** struggle particularly with adapting to the *Install* and *Google Apps* subsets, where task structures and requirements may differ significantly from the training set. This limitation suggests that these frameworks may be overfitting to the *General* task set or lacking the necessary mechanisms to capture the underlying transferable features essential for effective generalization.

The ability of DISTRL to generalize across diverse task subsets is a critical advantage, especially in real-world applications where agents are often required to operate in varied and unforeseen environments. This generalization strength is likely a result of the extensive exploration and varied experiences facilitated by the distributed learning process, which allows DISTRL to build a more comprehensive and flexible policy.

These findings have significant implications for the development of autonomous agents. The demonstrated generalization capabilities of DISTRL suggest that distributed RL frameworks can be a promising direction for creating agents that are not only proficient in specific tasks but also adaptable to a wide range of scenarios without the need for extensive retraining. Future research could investigate the integration of additional generalization techniques, such as meta-learning or multi-task learning, with distributed RL to further enhance performance across even more diverse and complex task domains.

### A.6.3 DISTRL AGENT PERFORMANCE ON TEST SET

Our primary contribution is improved training performance, which we evaluated using strictly separated training and testing sets. While our main results are reported in Section 6.5 using the final trained agent, we also tracked performance progression on test sets during training. Figure 13 shows the averaged results (3 trials) of periodic checkpoint evaluations on the "*General*" test set from AiTW, demonstrating continuous improvement throughout training.

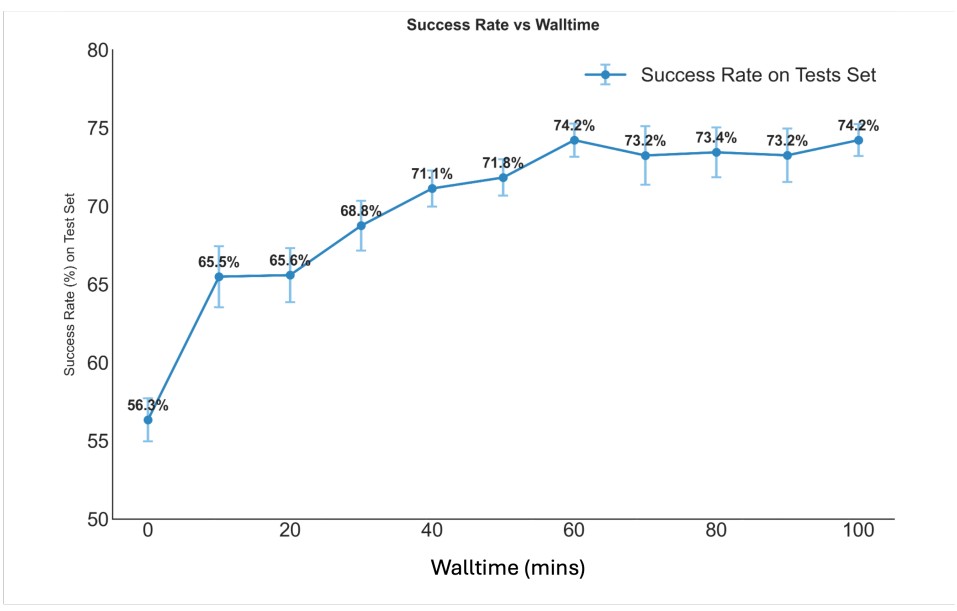

**Figure 13:** Continuous Evaluation on the Test Sets during DistRL fine-tuning process

### A.7 GENERALIZATIONS TO OTHER DOMAINS

**Expanding to a Broader Range of Mobile Applications**    We conducted additional experiments on "*GoogleApps*" and "*Install*" tasks from AiTW datasets, not included in the training set, to assess generalization. Results in Appendix A.6.2 demonstrate that DistRL maintains high performance and effectively generalizes to new tasks. However, achieving high generalization remains challenging when facing substantial variations across different apps. We are also exploring more open-world benchmarks, and as future work, we plan to expand our exploration to a wider range of mobile applications.

**Handling Significant UI Changes or App Updates**    DistRL's design accommodates different device types, OS versions, UI changes, and app updates by using ADB and emulators for human-like interactions and system resets. The main challenge is the compatibility of AVD and app versions. For major UI overhauls or new devices, additional training or fine-tuning may be required. However, DistRL's architecture supports scalable data collection and training across diverse environments, enhancing its robustness in real-world scenarios. We validate this robustness using our dedicated testing infrastructure, including proprietary OS environments and hardware.

**Adapting the Framework for Other OS**    To extend DistRL to iOS, we can adapt our framework by replacing Android-specific tools with Apple's counterparts (e.g., XCUITest, simctl) while maintaining our platform-agnostic core components. Our validation was facilitated by robust hardware and OS support that enabled efficient API resets and testing modules. The key challenge in migrating such RL fine-tuning platforms lies in the ability to reset states for trial-and-error learning. While our framework can support iOS environments with necessary adjustments, Android remains our primary platform due to its developer-friendly ecosystem.

