# OpenReview forum: "DistRL: An Asynchronous Distributed Reinforcement Learning Framework for On-Device Control Agent"
_ICLR.cc/2025/Conference — ICLR 2025 Poster_

### Official Review · Reviewer_oSkG · 2024-10-29

**Soundness:** 2
**Presentation:** 1
**Contribution:** 3
**Rating:** 8
**Confidence:** 4

**Summary:**

This paper focuses on the domain of on-device control agents and presents DistRL, an asynchronous distributed reinforcement learning framework for device control agents. This paper designs a centralized training and decentralized data acquisition system design to ensure the efficiency of online RL fine-tuning given potentially heterogenous hardware specifications. It also proposes a tailor-made RL algorithm in the context of distributed device control training. Both the system design and new RL algorithm shows improvements over the prior state-of-the-art DigiRL in a popular device-control benchmark AitW.

**Strengths:**

The paper addresses a realistic concern of asynchronomous data collection in applying RL fine-tuning for device-control agents, so it is highly relevant.

The proposed system-level optimizations can be valuable to the open-source community. I hope the authors can clean up and release the infrastructure for their experiments.

The proposed RL algorithm and system design seems to make a difference on popular AitW benchmark.

**Weaknesses:**

The writing of the paper, in particular the introduction and methodology, can be improved. In fact, the current state of writing has caused me trouble for understanding the contributions of the paper. A significant amount of re-writing is required before the paper can be published in a top-tier venue like ICLR. Please see my questions for more details.

The organization of the section 5 methodology makes it hard to understand what parts are novel in the paper and what parts are adapted from existing literature. From the current writing, it seems to be a combination of existing components from the prior literature. Additional motivations are needed to explain what is novel in the tailor-made RL algorithm.

**Questions:**

Line 53-70 seems repetitive.

Line 71 fine-tuning(type of light training) -> fine-tuning

Paragrapho one and two of intro seem to be a bit verbose in introducing the background.

Line 78-85, it seems unnecessary to introduce the notion of RLHF here cuz this paper does not have human preferences.

Line 86-90 is confusing, it seems to describe challenges of exisiting offline datasets such as AitW and Android-Control, but why is it relevant for online RL fine-tuning?

Line 98-104 can be best put in the related works.

There are too many bolded text in the introduction that can tire the readers. I would suggest removing the bold font for related methods. Same applies to the experiment section.

Line 121-122, te claim of "DistRL is the first work to scale autonomous, online RL fine-tuning for mobile device control in a distributed environment." may be incorrect, DigiRL also supports distributed data collection. I understand that DistRL supports more distributed RL optimizations, but it might be good to update this sentence.

Line 247-257, would be great to defer algorithm-related designs to the next section as this section mainly talks about system design.

There can be more details for the method. In particular, do you maintain a separate model for $Q$ and $\rho$? If so, how are those models trained? How is $\Pcal_{invalid}$ calculated? How is $A(s_t, a_t)$ calculated? More motivations can also be included. For example, why does DPER improve sample efficiency? 5.3 includes a training loss for the trajectory value function $V_{traj}$, where is it used?

Figure 5 (b) needs more explanations, what does "Proportion Above 60%" mean?

---

> ### Author Response · Authors · 2024-11-18
> **Response to Reviewer oSkG (Part 1)**
>
> We sincerely appreciate the reviewers' valuable feedback and have addressed each concern in detail below.
>
> ### Concern1: Paper writing
> We acknowledge all feedback regarding our paper's writing style and presentation. In our revised version, we have comprehensively improved the manuscript by:
>
> - Enhancing the clarity of the first two sections
> - Refining our use of bold text throughout the paper
> - Implementing all suggested writing improvements
>
> We are committed to delivering a more polished and professional manuscript that better communicates our research contributions.
>
> ### Concern2: Why did we say "DistRL is the first work to scale autonomous, online RL fine-tuning for mobile device control in a distributed environment."
>
> We appreciate the reviewer’s feedback and have worked to clarify and substantiate this claim in our revised version.
>
> Our claim that "DistRL is the first work to scale autonomous, online RL fine-tuning for mobile device control in a distributed environment" is grounded in extensive testing and rigorous reproduction of baseline methods. While DigiRL [3] nominally includes distributed data collection, it has significant limitations in terms of scalability. Our empirical analysis reveals that DigiRL’s implementation suffers from critical issues, including severe inefficiencies in training, incomplete implementation, resource contention when running multiple emulators, and ineffective use of distributed computing resources, all of which hinder its scalability for any **Online** deployments. In contrast, DistRL is explicitly designed with scalability in mind, incorporating optimizations for distributed architectures such as efficient resource allocation, load balancing, and robust training across multiple machines. This allows DistRL to provide a truly scalable solution for autonomous, online RL fine-tuning on mobile devices. The primary distinction lies not merely in having distributed functionality but in the effective and efficient use of distributed resources, enabling training stability and high performance at scale.
>
>
> ### Concern 3. Figure 5(b) needs more explanations; what does "Proportion Above 60%" mean?
>
> - The “Proportion Above 60%” refers to the proportion of recorded time steps within each interval where the agent achieves a success rate over 60%, with “Proportion Above 80%” indicating time steps exceeding 80%. These metrics highlight learning efficiency and stability. As shown in Figure 5(b), DistRL consistently achieves higher proportions than DigiRL[3], demonstrating faster convergence and improved stability due to its asynchronous architecture and optimized data collection and sampling strategies. We have clarified this in the figure caption and main text.

---

> ### Author Response · Authors · 2024-11-18
> **Response to Reviewer oSkG (Part 2)**
>
> ### Concern 4. Clarifications over method details of A-RIDE
>
> We have already added more details as suggested to Section 5 and Appendix A.4.
>
> - **Do you maintain a separate model for $Q$ and $\rho$? If so, how are those models trained?**
>
>
> 	Given your interpretation, since $\rho_t = \dfrac{\pi(a_t | s_t)}{\mu(a_t | s_t)}$ represents the importance sampling ratio between the target policy $\pi$ and the behavior policy $\mu$, we follow the idea that you may be asking about the relationship between the **value network** (critic) and the **policy network** (actor). Our framework employs separate models: a **ROBERTA-base** as the **value network** for estimating state values and a **policy network** for generating control actions.
>
>
> 	- The **value network**  estimates the state value function $V(s_t)$, representing the expected return from state $s_t$. Instead of directly regressing on scalar returns, we train the value network to predict the probability that the Monte Carlo return $G_t$ (including both immediate and future rewards) is positive. The critic’s objective is formulated as a binary classification task, with the following loss: $\mathcal{L}(V) = \mathbb{E} \left[ - \mathbb{I}[G_t > 0] \log V(s_t; \theta) - \left(1 - \mathbb{I}[G_t > 0]\right) \log \left(1 - V(s_t; \theta)\right) \right]$, where $V(s_t; \theta)$ is the predicted probability that $G_t > 0$, parameterized by $\theta$, $G_t = \sum_{k=t}^{H} \gamma^{k-t} r_k$ is the Monte Carlo return from timestep $t$, and $\mathbb{I}[G_t > 0]$ is an indicator function, which equals 1 if $G_t > 0$ and 0 otherwise. The value network parameters $\theta$ are optimized by $\theta^* = \arg\min_\theta \mathcal{L}(V; \theta)$
>
>
> 	- The **policy network** optimizes the following loss: $$\mathcal{L} = - \mathbb{E}\_{\mu} \left[ \rho_t A(s_t, a_t) \log \pi(a_t | s_t) \right] - \beta \mathbb{E}\_{\mu} \left[ \mathbb{H}(\pi) \right] + \lambda \mathbb{E}\_{\mu} \left[ \mathcal{P}\_{\text{invalid}} \right]$$
> 	  Here, $\rho_t$ corrects for off-policy data, $\mathbb{H}(\pi)$ encourages exploration, and $\mathcal{P}_{\text{invalid}}(a_t)$ penalizes invalid actions.
>
>
> - **How is $\mathcal{P}_{\text{invalid}}$ calculated?**
>
>     - The invalid action penalty term $\mathcal{P}_{\text{invalid}}$ penalizes actions that the agent cannot execute, addressing the challenge of entropy regularization, which encourages exploration but can lead to invalid actions. In mobile device control tasks, valid actions such as clicking and typing, while invalid commands, such as “rotate screen” in an unsupported context or interacting with non-existent UI elements, waste resources and hinder learning.
>
>       We use **Gemini-1.5-Pro** to evaluate each action $a_t$. If an action is invalid, $\mathcal{P}\_{\text{invalid}} = 1$; otherwise, it is 0. The penalty term is integrated into the loss as $\mathcal{L}\_{\text{penalty}} = \lambda \cdot \mathbb{E}[\mathcal{P}\_{\text{invalid}}]$, where $\lambda$ controls its impact. This approach ensures exploration remains within valid bounds, enhancing learning efficiency and robustness .
>
> - **How is $A(s_t, a_t)$ calculated?**
>
>     - The advantage function $A(s_t, a_t)$ is calculated using a one-step advantage estimation approach, as follows
>        $$
>        A(s_t, a_t) = r(s_t, a_t) + \gamma V(s_{t+1}) - V(s_t)
>        $$
>       Here, the formula comprises two future components: the Monte Carlo return from timestep $t+1$ to the terminal state $ s_H $, which is determined by success or failure signals, and hard-coded penalties for repeated actions and certain violations, serving as immediate reward signals. This combination provides a long-term cumulative reward estimate. $ V(s_t) $ represents the estimated state value at timestep $t$, $V(s_{t+1})$ is the estimated state value at the next timestep $ t+1 $, and $ \gamma $ is the discount factor.
>
>
> - **Why does DPER improve sample efficiency?**
>
>   - DPER improves sample efficiency by prioritizing high-value transitions, such as those with significant TD errors or high importance sampling ratios. This focus enables the agent to learn effectively from previously explored states, reducing unnecessary exploration of less relevant areas. By maintaining policy entropy, DPER balances exploration and exploitation, accelerating convergence with fewer samples.
>
> - **Section 5 includes a training loss for the trajectory value function $V_{\text{traj}}$, where is it used?**
>
> 	- The trajectory value function $V_{\text{traj}}$ estimates the expected return from terminal state $s_H$ in sparse/delayed reward environments. It serves as a baseline by labeling reward signals in collected trajectories (analogous to RLHF reward models) and filters/prioritizes trajectories for training. Through this filtering mechanism, $V_{\text{traj}}$ maintains high-quality training data, leading to more effective policy learning.
>
> **We sincerely invite you to consider raising your scores if our responses do address your concerns.**

---

> > ### Comment · Reviewer_oSkG · 2024-11-22
> >
> > Thanks for your rebuttal and the revisions in terms of the writing of the paper. My concerns have been adequately addressed.

---

### Official Review · Reviewer_w8MY · 2024-10-31

**Soundness:** 4
**Presentation:** 4
**Contribution:** 3
**Rating:** 8
**Confidence:** 4

**Summary:**

This paper provides a way to train device control agents using reinforcement learning. Specifically, it proposes a setup of asynchronous data collection using parallelized data collection on emulators (or real devices), combined with a centralized learning algorithm. The algorithm for training is V-trace, combined with prioritized replay buffer and an additional component encouraging exploration. The system runs on multiple workers and uses a single host learner.

The paper evaluates results on AitW using model-based evaluation. Compared to existing literature/SoTA the DistRL approach achieves a high success rate in less wall time by applying a distributed, asynchronous training approach, which results in faster emulation speed (traj/min). In the General test set, DistRL achieves a success rate a relative improvement of over 19% vs previous work (DigiRL) in the Web Shopping test set, DistRL attains a success rate  outperforming DigiRL by over 14%. The work conducts ablation analysis.

**Strengths:**

The work achieves impressive results on a challenging task. It provides a path to scale up data collection and enable future work to tackle even more complex problems. Implementing this distributed setup and making it work is a valuable contribution. This work is particularly relevant to future researchers where researchers can easily spin up many workers on cloud platforms.

The paper is well-written, with technical depth and there are enough details describing the methodology and approach. The paper does a good job of comparing their results to existing works and agents.

It is good the authors run multiple experiments per and average the results to provide more stable results. It is also good that the work includes an ablation study.

**Weaknesses:**

In order for this work to be a major contribution, I feel it should be open-sourced since a core contribution is the engineering work around centralized training and decentralized data acquisition, which requires complex communication between different systems. Open-sourcing will greatly help the community to take advantage of the contributions here and with reproducibility.

The core RL techniques used are relatively standard. There's no fundamental advance in machine learning theory or methodology. However combining IMPALA with prioritized experience replay seems to be novel, and it’s implementation is highly non-trivial.

If both DigiRL (single) and DigiRL (multi) have the same amount of emulators, why are the performance numbers between them different? Ultimately, both consist of 32 emulators, if I understand correctly, operating with synchronous training. Please clarify how they are different and why it's relevant to include both. I also think the number of emulators should be displayed more prominently as a key experimental parameter (rather than only in Figure 5)

Spelling/Writing: "ensuring that only contextually appropriate actions are penalized". I believe this should be “inappropriate actions are penalized”?

**Questions:**

“The evaluator receives the current observation, composed of the task description and a screenshot of the device, and outputs a reward signal.” – Have the authors validated that this does a good enough job of success validation, versus, say using the entire trajectory?

To me, “On-device End-to-End Agent Performance Evaluation” implies that the inference is done on the device, but this appears not to be the case. I understand it could be done in theory because the model is small enough, but it’s not actually on device as far as I can tell. Is there another way to describe it?

## Comments
(You don’t have to address these; they are intended to be helpful only.)

For future work, you may consider SeedRL to further scale the approach. https://arxiv.org/pdf/1910.06591#page=5.27

Future work could evaluate on online benchmarks such as AndroidWorld to ensure that results are easily comparable for future researchers, since model-based evaluation is harder to reproduce due to costs and because of differences in model outputs, even for the same base model.

---

> ### Author Response · Authors · 2024-11-18
> **Response to Reviewer w8MY**
>
> First of all, we really acknowledge the reviewer’s recognition over our works and address each concern in detail below.
>
> ### 1. Concerns regarding the performance discrepancy between DigiRL(single) and DigiRL(Multi):
>
> In the DigiRL (Single) setup, we strictly adhere to the original DigiRL baseline by running up to 32 emulators on a single host machine. However, we observed that implementation limitations prevent all emulators from operating simultaneously at full capacity, even on high-end hardware. This configuration employs synchronous training without optimization, leading to substantial resource contention and performance bottlenecks due to all emulators competing for the same computational resources. We use this setup as a baseline to highlight the limitations inherent in single-machine environments.
>
> In contrast, DigiRL (Multi) distributes the 32 emulators across multiple host machines, with each machine managing a manageable subset of emulators. While maintaining synchronous training, this approach effectively balances the computational load and incorporates minor performance optimizations. By leveraging a distributed architecture, DigiRL (Multi) mitigates the resource contention seen in the single-machine setup, resulting in better resource utilization and improved performance.
>
> ### 2. Validation of the VLM Evaluator (Gemini) Using the Entire Trajectory
>
> We appreciate the reviewer’s concern regarding the effectiveness of using the current observation versus the entire trajectory for success validation. In our experiments, we used the last screenshot along with the last two actions as input to the VLM Evaluator (Gemini) for prompting. This approach provides a concise yet informative context for the evaluator. Please see our supplementary experiments as we replied to Reviewer **vpis**, we tested the correlation between our autonomous evaluator and human judgments for different agents on the whole *General* subsets from AiTW:
>
> | Success Rate | AppAgent | | DigiRL |   | DistRL |   |
> |:---:|:---:|:---:|:---:|:---:|:---:|:---:|
> | | Human | Gemini | Human | Gemini | Human | Gemini |
> | X=1 | 42.3|    44.2       | 60.3   |   62.1   | **73.4**| **74.2**|
> | X=2 |42.3 | 44.7        |60.3  | 63.7 |73.4   | 75.5|
> | X=3 | 42.3| 48.2       | 60.3| 66.3	| 73.4   | 78.3|
>
>
> Our findings indicate that incorporating additional context, such as longer trajectory information, negatively impacts evaluation accuracy. Specifically, the discrepancy between the evaluator's output and human assessments was less than 2% when using only the last image and the last two actions. However, providing more context led to performance drops, significantly higher computational costs, and increased usage of evaluator LLMs, thereby straining our budget.
>
> In fact, empirical tests show that large models like Gemini tend to classify tasks as successful by identifying evidence of success in provided screenshots. The more information Gemini receives, the higher the probability it judges the task as successful. In few-shot scenarios with seven or eight images, adding more tokens causes token explosion, leading to hallucinations. Our algorithm leverages score assignments based on final steps and states, achieving a balance between computational efficiency and evaluation accuracy. We have updated the manuscript with these results and detailed analysis in section 6.3, demonstrating that Gemini provides reliable success validation with minimal input, which is essential for efficient and cost-effective training.
>
> ### 3. Clarification on "On-device End-to-End Agent Performance Evaluation"
>
> Thanks for highlighting the potential confusion in the section title. To avoid misinterpretation, we renamed the section to "Performance Evaluation of Agents Trained with DistRL." This new title more accurately reflects the content, focusing on evaluating the performance of agents trained using our distributed framework across various tasks and environments. We have also updated the text to clarify that while the agents are capable of running on-device due to their compact size, the evaluations were conducted using host machines with emulators for consistency and scalability.
>
> ### Additional Comments
> We are grateful for the reviewer’s suggestions regarding future work. We have dedicatedly revised the paper based on all these valuable comments. Exploring frameworks like SeedRL to further scale our approach is indeed a promising direction. Additionally, we acknowledge the value of evaluating our methods on online benchmarks such as AndroidWorld for standardized comparisons. We plan to incorporate these considerations in our future research to enhance the reproducibility and impact of our work.
>
> **We sincerely invite you to consider raising your score given your positive reviews.**

---

> > ### Comment · Reviewer_w8MY · 2024-11-22
> >
> > Thank you for the detailed response, including creating an anonymous OSS repo. I am glad to see this will be able to be broadly utilized by the community. My questions and concerns have been addressed.

---

### Official Review · Reviewer_fxPf · 2024-11-03

**Soundness:** 3
**Presentation:** 2
**Contribution:** 3
**Rating:** 5
**Confidence:** 3

**Summary:**

This paper introduces DistRL, an asynchronous distributed reinforcement learning (RL) framework designed to fine-tune on-device control agents, particularly for mobile environments. The key contributions include a scalable architecture that decouples data acquisition from policy learning, improving training efficiency and robustness in dynamic environments. The paper claims significant improvements in training speed and success rates compared to state-of-the-art methods, validated through experiments on Android tasks.

**Strengths:**

1. The decoupling of data acquisition from policy learning allows the framework to handle heterogeneous devices efficiently, which is crucial for real-world applications.
2. The asynchronous nature of DistRL enables faster data collection and policy updates compared to synchronous methods, which is demonstrated by a claimed 3x improvement in training speed.
3. The focus on mobile device control agents addresses a practical problem with significant industry relevance, particularly in automating complex user interactions on smartphones.

**Weaknesses:**

1.  There is a significant discrepancy between the model descriptions in different sections (T5-based multimodal model mentioned in Sec 1 vs GPT-2/ROBERTA in A.5.1). This inconsistency undermines the paper’s credibility and makes it difficult to understand what model was actually used. If different models were used for different tasks or components (e.g., policy learning vs critic evaluation), the authors should explain why and provide a rationale for these choices. If there were intentional changes in model usage (e.g., using GPT-2 for policy learning and ROBERTA for critic evaluation), the authors should explain why these variations were necessary and how they impacted performance.

2. It is unclear how the proposed method differs from IMPALA from the methodology perspective. I understand that applying them to a new on-device setting is trivial, but it is confusing to me what are the major differences from the algorithm side. It seems that A-RIDE builds upon existing methods like Retrace and GAE but does not introduce substantial innovations beyond adapting these techniques to a distributed setting. Additionally, the author mentioned in Sec. 2.3 that:
> Furthermore, we develop a novel reinforcement learning algorithm specifically designed to handle the stochasticity and scalability inherent in mobile device environments.
However, I did not see any specific techniques and algorithm improvements that address the stochasticity and scalability issues.
If there are indeed major algorithm changes, it would be beneficial to add a baseline of IMPALA and IMPACT.

3.  Additionally, the training and testing set used are not clearly described. I am a bit confused about the difference between the training and testing sets -- are there overlaps of tasks between them? How is the expert-curated and AndroidWorld training set selected? This questions the generalization issues of the experiments.

4. The evaluation metrics and result plots are a bit confusing. It is unclear whether the plots in Fig. 6 are for the training task success rate or the testing task's. Additionally, adding the testing task success rate plots instead of just training one in Fig. 5 would be beneficial as well.

5. The x-axis in Fig. 5 (a) is the time. From Fig. 5 (c) we can see there are significantly more trajectories collected than baselines. I acknowledge that the implementations of the proposed method is much more efficient than baselines, but I am also curious about what the plot looks like if the x-axis is the trajectory-num or the step-num, which is commonly reported in the RL literature. It would be super helpful to understand the upper bounds of the baseline algorithms given the same number of collected data. It is also interesting to know the synchronized algorithm performance trend.

**Questions:**

See weakness

---

> ### Author Response · Authors · 2024-11-18
> **Response to Reviewer fxPf (Q1)**
>
> We greatly appreciate the reviewers' insightful suggestions and helpful questions, which will enhance our paper's quality. We address each question individually below:
>
> ### Q1: Clarification on Model Usage and Implementation Details
>
> In practice, we did not encounter any conflicts during the implementation of our system, and there were no intentional changes in model usage.
>
> In our system’s implementation, we maintained a consistent and conflict-free approach by exclusively utilizing the T5-based Multimodal Large Language Model (MLLM) architecture (not vanilla pretrained T5, T5-Base normally has 220M, Our Agent (AutoUI-driven) has 1.3B). This encoder-decoder framework facilitates the seamless integration of various agents, enhancing the model’s versatility and performance, which is aligned with AutoUI [42] design.
>
> Our T5-based MLLM architecture is inherently designed to support diverse decoder initializations, including the integration of pretrained decoder models. While previous works like AutoUI [42] have utilized their own model weights, we could not directly use AutoUI’s model weights to initialize our models because they were trained with AiTW knowledge. Incorporating these weights could introduce unintended biases specific to their tasks as well as prior knowledge to our training and testing data set, which might reduce the credibility and confidence of our system.
>
> Meanwhile, our empirical analysis showed that decoder layers require more diverse initialization patterns than encoder layers to achieve optimal performance in UI-specific tasks, GPT-2 help more than the pre-trained T5 weights. Therefore, the best approach was to manually select a pretrained decoder model to initialize the weights of our T5-based decoder. We chose to initialize the decoder with GPT-2 weights, as this allowed us to leverage GPT-2’s pre-trained language generation capabilities, providing a foundational understanding of language that we could further refine through fine-tuning tailored to our specific tasks.
>
> We did not specify the decoder type/weights in the original explanation because we found that fine-tuning efficiency was already satisfactory with GPT-2. We acknowledge that there might be better options available; however, we adopted this approach simply by drawing inspiration from our baseline DigiRL [3], which also used GPT-2 to initialize their decoders (This is a *trick* here). By following a structured and methodical approach from existing baselines, we ensured architectural compatibility and effective integration of the GPT-2 weights into our T5-based decoder.
>
> For clarity, refer to our submitted code: `https://anonymous.4open.science/r/DistRL-C41F/distrl/models`, particularly `./models/model1.py` lines 16-30 and `./models/autoui_agent.py` lines 20-54. We apologize for any misunderstandings caused by our statements and have already made the statements clearer in the revised version.

---

> ### Author Response · Authors · 2024-11-18
> **Response to Reviewer fxPf (Q2)**
>
> ## Q2: Concerns Regarding Novelty
> We have revised **Section 2.3** and **Section 5** to ensure the organization is excellent and the coverage is thorough.
>
> ## A. Concerns Regarding IMPALA and IMPACT
>
> As discussed in section 2.3, though old, IMPALA and IMPACT are well-known distributed reinforcement learning algorithms; however, the original work were not directly used as baselines to several key reasons:
>
> 1. **Inadequate Handling of Fluctuating Online Experiences:** Both IMPALA and IMPACT were designed with environments in mind that are relatively stable and do not exhibit the high variability found in mobile device interactions.
> 2.  **Lack of efficient buffer management for highly distributed cases.**
> 3. **Less supports from off-policy RL Models.** It leads to very **poor** training performance.
> 4. **Scalability and System Design Differences:** IMPALA and IMPACT do not offer the same level of scalability and system optimizations required for distributed mobile control.
>
> However, though non-trivial, effective implementation **upon** IMPALA is doable, here comes our work. Specifically, managing communication, scheduling worker roles, and handling queues and replay buffers demand substantial effort and innovative design solutions, e.g.:
>
>    - split FIFO queues into chunks and build circular buffers to support distributed use.
>    - Resolve thread and process locks and conflicts.
>    - Optimize memory usage of buffers.
>    - tailored off-policy RL as mentioned below
>
> ## B.	Concerns Regarding Algorithm Novelty
>
> Regarding the algorithm on the learner side, we believe that "making things usable is more important than making things fancy" in terms of overall system operations. Our tailored RL model aims to solve two main problems:
>
> - (A) Instabilities from distributed online learning.
> - (B) Efficient utilization of experiences collected from different time spans across workers.
>
> We address (A) using **ReTrace** and (B) using **DPER** , whose individual effects are validated in our ablation studies in Section 6.6 and Figure 7(b).
>
> Beyond the design of DPER and ReTrace, our work builds upon the foundation of Generalized Advantage Estimation (GAE) and DigiRL [3] to address the challenges inherent in distributed, asynchronous reinforcement learning environments with communication delays through two key innovations. See the following details:
>
>
> ### **Foundation and Enhancement of GAE**:
> - **Vanilla GAE** estimates advantages using TD errors across multiple timesteps:
> $A^{\text{GAE}}(s_t, a_t) = \sum_{l=0}^{\infty} (\gamma\lambda)^l \delta_{t+l}$
>
> - **The difference compared with GAE is the use of value estimation to estimate advantage.** Firstly, we introduces trajectory-level value estimation to label trajs, i.e.,  a trajectory-level value estimator $ V_{\text{traj}} $  (parameterized by Network $\theta$). 	where $V_{\text{traj}}$ is trained to capture long-term dependencies and trained with MLE loss:
>
> $$ \arg\mathop{\min}\limits_{\theta} \mathcal{L}(V_{\text{traj}}) = -\mathbb{E}\_\nu [ r(s_H, a_H) \log V_{\text{traj}}(s_H, a_H) + (1-r(s_H, a_H)) \log(1-V_{\text{traj}}(s_H, a_H)) ]$$
>
>    Trained trajectory critic serves dual purposes: labeling rewards and filtering the replay buffer to retain only high-value trajectories.
>
>    Given that our primary signals are extended and delayed rewards, we then train a Value-network $V(s_t)$ to estimate expected cumulative rewards directly from states rather than learning $Q$ values via trajs' success or failure signals from MC-Return. The advantage function is finally computed using the one-step formulation $A(s_t,a_t) = r(s_t,a_t) + \gamma V(s_{t+1}) - V(s_t)$, where $r(s_t,a_t)$ includes some hard-coded intermediate rewards/penalties.
>
> It's worth mentioning that in our practice, we have off-policy corrections using **ReTrace** upon the value function, please see the **Section 5.1-5.2** in the revised version.
>
>
>
> ### **Policy Optimization with Robust Regularization**:
> Once we have obtained advantage function $A(s_t,a_t)$ and we can train the actor based on the  policy loss
>
> - We extend Digirl's advantage-weighted objective:
> $\mathcal{L}\_{\text{AWR}} = -\mathbb{E}\_{\nu}[\log \pi(a|s) \cdot \exp(A(s,a)/\beta)]$ to a more robust formulation with importance sampling and regularization:
> $\mathcal{L} = -\mathbb{E}\_{\mu} [ \rho_t A(s_t, a_t) \log \pi(a_t | s_t) ] - \beta \mathbb{E}\_{\mu} [ \mathbb{H}(\pi) ] + \lambda \mathbb{E}\_{\mu} [ \mathcal{P}_{\text{invalid}} ]$
>
> This formulation improves upon both GAE and [3] by:
>
> - Incorporating importance sampling ($\rho_t$) for stable off-policy learning
> - Adding entropy regularization and invalid action penalties for robust exploration
>
>
> Since we did not position our work as a novel algorithm-centric paper, we did not dedicate extensive space to this explicit policy learning part and only mentioned the core designs. We have added these details to the revised version to enhance clarity.

---

> ### Author Response · Authors · 2024-11-18
> **Response to Reviewer fxPf (Q3-Q5)**
>
> ### Q3: Clarification on Training and Testing Sets, and Generalization Issues
>
> We ensured no overlap between training and testing sets by proper splitting during evaluations. Training sets were selected and formed based on empirical results and iterative trial-and-error, focusing on areas where pure AiTW is limited. For example:
>
>  - AiTW tends to result in Chrome-based operations.
>  - AndroidWorld includes general app controls like "create contact number."
>  - Expert-added tasks involve more detailed operations, such as checking specific app updates in the app store.
>
>
> It’s worth mentioning that we did not use any traced data from AiTW, only their task instructions. Training and testing sets examples are provided in Appendix A.5.2, Tables 4 and 5. We will open-source these sets post-review.
>
> ### Q4: Performance Evaluation During Training and End-to-End Training Considerations
>
> Our primary contribution is improved training performance. We evaluated the trained DistRL agent independently with no overlaps between training and testing sets. The only performance evaluation we conducted was after the training period; we evaluated the trained agent produced by DistRL independently, as reported in Section 6.5.
> However, we are pleased to provide additional results as requested. During training, we can frequently evaluate checkpoints on the test set. Results averaged in 3 trials are presented here (tested on “General” from AiTW as example):
>
> | Walltime (mins) | Avg SR-testset | Std |
> |:---:|:---:|:---:|
> | 0 | 56.34% | ±1.5% |
> | 10 | 65.49% | ±1.2% |
> | 20 | 65.59% | ±1.2% |
> | 30 | 68.75% | ±1.0% |
> | 40 | 71.13% | ±0.8% |
> | 50 | 71.83% | ±0.9% |
> | 60 | 74.22% | ±0.8% |
> | 70 | 73.24% | ±1.2% |
> | 80 | 73.44% | ±1.0% |
> | 90 | 73.25% | ±1.0% |
> | 100 | 74.22% | ±0.8% |
>
>
> These additional results with more details were reported in the revised version in Appendix A.6.3 to offer more insights into the agent's performance over time. The reviewer can also check here: `https://anonymous.4open.science/r/DistRL-C41F/Rebuttal_extra_exps/TestCurve-Rebuttal.png`. We have also included these results in Appendix A.6.3 in the revised paper.
>
> ### Q5: Suggestion to Plot Performance Curves Against the Number of Environment Interactions Instead of Time
>
> We appreciate the suggestion; however, in distributed RL systems, plotting against time is standard due to asynchronous processes (e.g., IMPALA [6], Figure 6). In fact, comparing on a 'per step' basis may not provide the most balanced perspective in this context because:
>
>  - **DigiRL-Multi operates in a purely synchronous setting.** For each model training step on the learner side, it waits for a fixed number of collected trajectories. This results in a trend that is a straight line with the same scale as the buffer size.
>  - **Our design** collects trajectories much faster than the baselines but with definitely fewer trajectories collected "per step" as measured by the learned models.
>
> We also provided experimental results as per the reviewer's request within the same time budgets here:`https://anonymous.4open.science/r/DistRL-C41F/Rebuttal_extra_exps/Figure5(c)-Rebuttal.pdf`. These show that, given the same time, our system achieves higher collection speeds while maintaining input training information.
>
> **We sincerely invite you to consider raising your scores if our responses do address your concerns**

---

> ### Author Response · Authors · 2024-11-24
> **We appreciate any feedbacks from Reviewer**
>
> Dear Reviewer fxPf,
>
> Thank you sincerely for your valuable feedback on our paper.
>
> As the discussion period comes to a close, we hope our additional experiments and thorough clarifications have satisfactorily addressed your concerns. We have dedicated significant effort to implement all your suggestions and provide an in-depth analysis.
>
> We sincerely invite you to consider raising your scores if our responses have effectively addressed your concerns. Please let us know if there's anything further you'd like us to address—we are more than happy to continue the discussion. **Your perspective is highly important to us, and we look forward to your response.**

---

### Official Review · Reviewer_vpis · 2024-11-04

**Soundness:** 3
**Presentation:** 3
**Contribution:** 3
**Rating:** 6
**Confidence:** 4

**Summary:**

The paper presents DistRL, an asynchronous distributed reinforcement learning framework designed explicitly for on-device control agents in mobile environments. DistRL aims to optimize training efficiency and agent performance for complex tasks, leveraging centralized training with decentralized data acquisition. By introducing a unique off-policy algorithm, A-RIDE, DistRL effectively manages asynchronous data collection and prioritizes valuable interactions, addressing challenges like delayed updates and non-stationary data. Experiments demonstrate its significant performance and efficiency improvements over existing frameworks, especially for real-world mobile tasks, positioning DistRL as a scalable solution in distributed reinforcement learning for mobile devices.

**Strengths:**

- DistRL employs a decoupled architecture that enhances scalability and robustness by distributing data acquisition across heterogeneous mobile devices, making it efficient for large-scale deployment. Moreover, they propose an A-RIDE algorithm adapted for asynchronous environments, balancing exploration and stability, which improves sample efficiency and task success rates.
- The paper provides comprehensive experiment results that significantly outperform baselines, showcasing superior success rates in complex tasks. Ablation studies highlight the importance of its components, like DPER and retrace correction.
- The paper is well-written and easy to follow.

**Weaknesses:**

- Lots of hyperparameters are required. (e.g., Equation 1, 2, DPER). Do you have any tips or evidence for determining them, and are the values sensitive to different types of environments?
- Although the experiments cover general tasks, expanding the testing to a broader range of mobile applications would offer more substantial evidence of DistRL’s adaptability. (from additional tasks from AITW and AndroidWorld)
- Is VLM Evaluator (Gemini)'s performance in providing reward signals credible? Did you encounter any failure cases where VLM misleadingly provides success/failure signals? AndroidWorld provides manually defined dense rewards. Have you tried training DistRL with such a reward?
- While the asynchronous design improves adaptability, more detail on DistRL's response to drastic changes in mobile environments or new, unseen tasks would strengthen the discussion of its robustness in dynamic settings.

**Questions:**

- Any results from INSTALL and GOOGLEAPPS of AITW? / Why not compare results in AndroidWorld tasks?
- In real-world cases, many users (workers) have different types of devices and different OS versions. How does DistRL handle scenarios where the mobile device encounters significant UI changes or unexpected app updates during training?
- What are the trade-offs between computational overhead and empirical performance (success rate) when using more environments (workers) and Distributed Prioritized Experience Replay (DPER)?
- Could the framework be adapted for other operating systems, such as iOS, and if so, what modifications would be necessary?

---

> ### Author Response · Authors · 2024-11-18
> **Response to Reviewer vpis**
>
> We sincerely appreciate the reviewers' valuable feedback and answer as below:
>
> ## A. Main Weaknesses and Concerns
> **1. Hyperparameter Sensitivity and Determination**
>
> While our approach involves multiple hyperparameters, we effectively manage them through domain heuristics to constrain the parameter space and search. Detailed hyperparameter configurations and tuning procedures are documented in Appendix A.4.3.
>
> **2. Expanding to a Broader Range of Mobile Applications**
>
> We conducted additional experiments on "GoogleApps" and "Install" tasks from AiTW datasets, not included in the training set, to assess generalization. Results in Appendix A.6.2 demonstrate that DistRL maintains high performance and effectively generalizes to new tasks. However, achieving high generalization remains challenging when facing substantial variations across different apps. We are also exploring more open-world benchmarks, and as future work, we plan to expand our exploration to a wider range of mobile applications.
>
> **3. Credibility of our evaluator**
>
> To validate Gemini's reliability (According to our rewarding design, we only need to justify whether the task is successful or not), we conducted supplementary experiments comparing its evaluations with human assessments, i.e., we tested correlation between our autonomous evaluator and human judgements agents on whole
> *General* subsets from AiTW:
>
> | Success Rate | AppAgent | | DigiRL |   | DistRL |   |
> |:---:|:---:|:---:|:---:|:---:|:---:|:---:|
> | | Human | Gemini | Human | Gemini | Human | Gemini |
> | X=1 | 42.3|    44.2       | 60.3   |   62.1   | **73.4**| **74.2**|
> | X=2 |42.3 | 44.7        |60.3  | 63.7 |73.4   | 75.5|
> | X=3 | 42.3| 48.2       | 60.3| 66.3	| 73.4   | 78.3|
>
> X is used to represent evaluation(rewarding) via **Last X images**
>
>
> The analysis showed a high correlation with discrepancies in less than 2% of cases, demonstrating Gemini as a reliable proxy that reduces tons of manual labeling costs. Since rewarding from external evaluator is very close to the human labelling, we didn’t introduce other rewarding mechanism in our current research. But I believe using dense rewards will be a good future direction. (More analysis can be found in my responses to Reviewer **w8MY**)
>
> ## B. Answers to Questions
>
> **Q1. Results from Other AiTW tasks and Comparison with AndroidWorld**
>
> We clarify that we used the same task set as prior work and focused our testing on **AiTW** tasks to enable direct comparisons with existing baselines, e.g., DigiRL [3]. While other AiTW tasks would indeed be broader, they are not suitable for scientific projects (e.g. tasks that involve accounts) or have a very slow response time (e.g. App installation). However, we have also performed experiments on the "INSTALL" and "GOOGLEAPPS" tasks from AiTW to validate generalization abilities. The results, which are included in Appendix A.6.2, Figure 11, indicate that DistRL performs well on these additional tasks. (We didn’t exclusively train our agents on these tasks but treated them as generalization test sets)
>
> **Q2. Handling Significant Changes**
>
> DistRL’s design accommodates different device types, OS versions, UI changes, and app updates by using ADB and emulators for human-like interactions and system resets. The main challenge is the compatibility of AVD and app versions. For major UI overhauls or new devices, additional training or fine-tuning may be required. However, DistRL’s architecture supports scalable data collection and training across diverse environments, enhancing its robustness in real-world scenarios. We validate this robustness using our dedicated testing infrastructure, including proprietary OS environments and hardware.
>
> **Q3.Computational Overhead and Empirical Performance trade-off**
>
> In our evaluation of the computational overhead associated with using more environments (workers) and implementing DPER, we found the overhead to be minimal compared to the performance gains achieved. Specifically, we conducted some supplementary experiments as below:
> Evaluating computational overhead with 96 vCPUs and 32 emulators, DPER introduced ~7% training overhead compared to without DPER, while improving the average success rate from 5.3% to 13.3% (compared with baselines [3]), which is significant in this context. Optimization techniques like caching, distributed sampling, and priority updates every 5 steps minimized overhead. Communication overhead among scaling workers and learners is managed efficiently, with plans to utilize high-speed data transfer technologies like InfiniBand or RoCE.
>
> **Q4. Adapting the Framework for Other OS**
>
> To extend DistRL to iOS, trajectory input and environment resets would utilize Apple’s tools (e.g., XCUITest, simctl) instead of Android-specific tools. The core components are platform-agnostic, and with necessary adjustments, DistRL can support iOS environments. However, Android is still the first choice now due to the developer-friendly environment.

---

> > ### Comment · Reviewer_vpis · 2024-11-22
> >
> > Thanks for the detailed responses and additional results. Most of my concerns are well addressed, and I will keep the original score.

---

### Author Response · Authors · 2024-11-18
**Overview of Our Response & Open Source Concerns**

We sincerely appreciate all the valuable suggestions and acknowledge the reviewers' agreement that our contributions are significant within the application domain. We aim to present our work as a cohesive and integrated system for everyone's use. **Our work is driven by real-world practice and substantial needs for developing efficient, asynchronous RL fine-tuning systems for LLM agents**.

## **Commitment to Open Source and Transparency**

**In adherence to anonymity policies, we are committed to releasing the complete codebase, model weights, necessary prompts, datasets, and product demos for DistRL post reviews shortly.**

Interested parties and reviewers can refer to the temporary code repository here: `https://anonymous.4open.science/r/DistRL-C41F/`.

## **Revisions Based on Valuable Feedback**

We have thoroughly revised our paper to incorporate all the insightful comments received (**Main revisions were marked with titles highlighted in blue**). The main revisions include, but are not limited to:

1. **Additional Experiments:**
	We have included the requested extra experiments in both the appendix and the main body of the paper, particularly focusing on the validation results of our evaluators (See Section 6.3).

2. **Broader Discussions on Generalization:**
	Expanded discussions on the broader scopes of generalization abilities across various applications have been added as per the reviewers' suggestions (See Section A.6.2, A.7).

3. **Enhanced Methodology Section**
The methodology section (Section 5) has been revised to emphasize our novel algorithm in comparison to existing works. Additional details have been provided in Section A.4 to clarify our approach further.

4. **Improved Presentation and Corrections**
We have meticulously corrected typos, replaced citations with concise forms, removed redundant bold texts for better readability, and enhanced the overall presentation quality of the paper. (Mainly from Section 1 to Section 2)

Some references I mentioned in the responses below:

 - [3] Hao Bai, Yifei Zhou, Mert Cemri, Jiayi Pan, Alane Suhr, Sergey Levine, and Aviral Kumar. Digirl: Training in-the-wild device-control agents with autonomous reinforcement learning. arXiv preprint arXiv:2406.11896, 2024.

 - [6] Lasse Espeholt, Hubert Soyer, Remi Munos, Karen Simonyan, Vlad Mnih, Tom Ward, Yotam Doron, Vlad Firoiu, Tim Harley, Iain Dunning, et al. Impala: Scalable distributed deep-rl with im- portance weighted actor-learner architectures. In International conference on machine learning, pp. 1407–1416. PMLR, 2018.

 - [42] Zhuosheng Zhan and Aston Zhang. You only look at screens: Multimodal chain-of-action agents. arXiv preprint arXiv:2309.11436, 2023.

We believe these revisions address the reviewers' concerns comprehensively and significantly improve the quality and clarity of our work.

---

### Author Response · Authors · 2024-12-04
**Global Response - Discussion Summary**

Dear PC, SAC, AC, and Reviewers,

We sincerely appreciate your valuable suggestions and are grateful for the consensus among the reviewers regarding the significance of our contributions within the application domain. Our goal is to present our work as a cohesive and integrated system for widespread use, driven by real-world practices and the substantial need for developing efficient, distributed, and asynchronous reinforcement learning (RL) fine-tuning frameworks for large language model (LLM) agents on mobile devices.

After extensive discussions, we have addressed most of the concerns raised by the reviewers. Reviewers **vpis, oSkG, and w8MY** have expressed a very positive attitude toward our work, acknowledging its strengths, including its novelty, strong performance, and comprehensive experiments and ablation studies, as recognized by all reviewers.

Possibly due to the busy schedule, reviewer **fxPf** did not provide feedback on our responses, in which we have provided detailed experiments and responses. However, we believe they can resolve the issues raised by Reviewer **fxPf**.


**Commitment to Open Source and Transparency**

In adherence to anonymity policies, we are committed to releasing the complete codebase, model weights, necessary prompts, datasets, and product demonstrations for DistRL shortly after the review process concludes.

We extend our heartfelt thanks to the reviewers for their thorough evaluations and invaluable feedback throughout the discussion period. We greatly appreciate the recognition and thoughtful assessment of our paper.


Best regards,
Authors

---

### Meta-Review · Area_Chair_5M73 · 2024-12-19

**Metareview:**

**summary**

The paper introduces DistRL, an asynchronous distributed RL framework designed for training on-device control agents in mobile environments. Combining centralized training with decentralized data collection, DistRL addresses challenges like delayed updates and non-stationary data through its novel off-policy algorithm, A-RIDE, which incorporates asynchronous data collection, prioritized replay, and exploration-driven components. The scalable architecture decouples policy learning from data collection, enhancing training efficiency and robustness across heterogeneous hardware. Evaluations on the AitW benchmark demonstrate significant improvements over the state-of-the-art DigiRL, achieving faster training, higher emulation speeds, and success rate improvements of over 19% in the General test set and 14% in the Web Shopping test set.

---

**strengths**

* Novel idea of decoupling architecture and A-RIDE algorithm for asynchronous environments.
* Comprehensive results: Experimental results demonstrate significant performance improvements over baselines, showcasing superior success rates in complex tasks. Also, the asynchronous nature of the framework accelerates data collection and policy updates, achieving a 3x improvement in training speed compared to synchronous methods.
* Practical topic: The framework addresses a real-world problem of mobile device control, focusing on automating complex user interactions with high industry relevance.
* The paper is well-written, providing detailed descriptions of the methodology and clear comparisons with existing works.

---

**weaknesses**

* Evaluation ambiguities: The evaluation metrics and result plots are somewhat unclear. The reliance on VLM Evaluator (Gemini) for reward signals raises questions about its accuracy. The paper does not discuss potential failure cases where VLM may provide misleading success/failure signals.
* Given that a significant contribution of the work lies in its engineering, including centralized training and decentralized data acquisition, open-sourcing the implementation would enhance reproducibility and provide value to the community.
* Incremental innovation: The RL techniques employed, such as combining IMPALA with prioritized experience replay, are relatively standard and do not represent a fundamental advance in machine learning theory. However, their implementation in a distributed setting is novel and non-trivial.

---

**decision**

I think the paper makes a nice contribution that the community will find valuable. However, I encourage the authors to think carefully about how to reflect on the comments from reviewers (+open-sourcing the framework) in the camera-ready version.

**Additional Comments On Reviewer Discussion:**

The authors effectively addressed concerns raised by the reviewers, including evaluation metrics, experimental setups, and open-sourcing, during the discussion period, and most reviewers expressed satisfaction with their responses.

---

### Decision · Program_Chairs · 2025-01-22

Accept (Poster)